# EMBODIED ACTIVE DEFENSE: LEVERAGING RECURRENT FEEDBACK TO COUNTER ADVERSARIAL PATCHES

**Lingxuan Wu**[1], **Xiao Yang**[1*], **Yinpeng Dong**[1,2], **Liuwei Xie**[1], **Hang Su**[1], **Jun Zhu**[1,2*]
[1] Dept. of Comp. Sci. and Tech., Institute for AI, Tsinghua-Bosch Joint ML Center, THBI Lab, BNRist Center, Tsinghua University, Beijing, 100084, China    [2] RealAI
{wlx23,yangxiao19}@mails.tsinghua.edu.cn, lxiear@outlook.com
{dongyinpeng, suhangss, dcszj}@mail.tsinghua.edu.cn

## ABSTRACT

The vulnerability of deep neural networks to adversarial patches has motivated numerous defense strategies for boosting model robustness. However, the prevailing defenses depend on single observation or pre-established adversary information to counter adversarial patches, often failing to be confronted with unseen or adaptive adversarial attacks and easily exhibiting unsatisfying performance in dynamic 3D environments. Inspired by active human perception and recurrent feedback mechanisms, we develop Embodied Active Defense (EAD), a proactive defensive strategy that actively contextualizes environmental information to address misaligned adversarial patches in 3D real-world settings. To achieve this, EAD develops two central recurrent sub-modules, *i.e.*, a perception module and a policy module, to implement two critical functions of active vision. These models recurrently process a series of beliefs and observations, facilitating progressive refinement of their comprehension of the target object and enabling the development of strategic actions to counter adversarial patches in 3D environments. To optimize learning efficiency, we incorporate a differentiable approximation of environmental dynamics and deploy patches that are agnostic to the adversary's strategies. Extensive experiments demonstrate that EAD substantially enhances robustness against a variety of patches within just a few steps through its action policy in safety-critical tasks (*e.g.*, face recognition and object detection), without compromising standard accuracy. Furthermore, due to the attack-agnostic characteristic, EAD facilitates excellent generalization to unseen attacks, diminishing the averaged attack success rate by $95\%$ across a range of unseen adversarial attacks.

## 1 INTRODUCTION

Adversarial patches (Brown et al., 2017) have emerged as a prominent threat to the security of deep neural networks (DNNs) in prevailing visual tasks (Sharif et al., 2016; Thys et al., 2019; Xu et al., 2020; Zhu et al., 2023). These crafted adversarial patches can be maliciously placed on objects within a scene, aiming to induce erroneous model predictions in real-world 3D physical environments. As a result, this poses security risks or serious consequences in numerous safety-critical applications, such as identity verification (Sharif et al., 2016; Yang et al., 2022; Xiao et al., 2021), autonomous driving (Song et al., 2018; Zhu et al., 2023) and security surveillance (Thys et al., 2019; Xu et al., 2020).

Due to the threats, a multitude of defense strategies have been devised to boost the robustness of DNNs. Adversarial training (Madry et al., 2017; Wu et al., 2019; Rao et al., 2020) is one effective countermeasure (Gowal et al., 2021) that incorporates adversarial examples into the training data batch. Besides, input preprocessing techniques like adversarial purification aim to eliminate these perturbations (Xiang et al., 2021; Liu et al., 2022; Xu et al., 2023). Overall, these strategies predominantly serve as **passive defenses**; they mitigate adversarial effects on uncertain monocular observations (Smith & Gal, 2018) with prior knowledge of the adversary (Naseer et al., 2019; Liu et al., 2022; Xu et al., 2023). However, passive defenses possess inherent limitations. First, they remain susceptible to unseen or adaptive attacks (Athalye et al., 2018a; Tramer et al., 2020) that evolve to circumvent the robustness, owing to their dependence on presuppositions regarding the

---

*X. Yang and J. Zhu are corresponding authors.

adversary's capabilities. Second, these strategies treat each static 2D image independently without considering the intrinsic physical context and corresponding understanding of the scene and objects in the 3D realm, potentially rendering them less effective in real-world 3D physical environments.

As a comparison, human perception employs extensive scene precedents and spatial reasoning to discern elements that are anomalous or incongruent. Some research (Thomas, 1999; Elsayed et al., 2018) illustrates that human perception can effortlessly pinpoint misplaced patches or objects within 3D environments, even when such discrepancies trigger errors in DNNs on individual viewpoints. Inspired by this, we introduce **Embodied Active Defense (EAD)**, a novel defensive framework that actively contextualizes environmental context and harnesses shared scene attributes to address misaligned adversarial patches in 3D real-world settings. By separately implementing two critical functions of active vision, namely perception and movement, EAD comprises of two primary sub-modules: the perception model and the policy model. The perception model continually refines its understanding of the scene based on both current and past observations. The policy model, in turn, derives strategic actions based on this understanding, facilitating more effective observation collection. Working in tandem, these modules enable EAD to actively improve its scene comprehension through proactive movements and iterative predictions, ultimately mitigating the detrimental effects of adversarial patches. Furthermore, our theoretical analysis also validates the effectiveness of EAD in minimizing uncertainties related to target objects.

However, training an embodied model designed for environmental interactions presents inherent challenges, since the policy and perception models are interconnected through complex and probabilistic environmental dynamics. To tackle this, we employ a deterministic and differentiable approximation of the environment, bridging the gap between the two sub-modules and leveraging advancements in supervised learning. Moreover, to fully investigate the intrinsic physical context of the scene and objects, we deploy adversary-agnostic patches from the Uniform Superset Approximation for Adversarial Patches (USAP). The USAP provides computationally efficient surrogates encompassing diverse potential adversarial patches, thus precluding the overfitting to a specific adversary pattern.

Extensive experiments validate that EAD possesses several distinct advantages over typical passive defenses. First, in terms of **effectiveness**, EAD dramatically improves defenses against adversarial patches by a substantial margin over state-of-the-art defense methods within a few steps. Notably, it maintains or even improves standard accuracy due to instructive information optimal for perceiving target objects in dynamic 3D environments. Second, the attack-agnostic designs allow for exceptional **generalizability** in a wide array of previously unseen adversarial attacks, outperforming state-of-the-art defense strategy, by achieving an attack success rate reduction of $95\%$ across a large spectrum of patches crafted with diverse unseen adversarial attacks. Our contributions are as follows:

- To our knowledge, this work represents the inaugural effort to address adversarial robustness within the context of embodied active defense. Through theoretical analysis, EAD can greedily utilize recurrent feedback to alleviate the uncertainty induced by adversarial patches in 3D environments.
- To facilitate efficient EAD learning within the stochastic environment, we employ a deterministic and differentiable environmental approximation along with adversary-agnostic patches from USAP, thus enabling the effective application of supervised learning techniques.
- Through exhaustive evaluations, we demonstrate that our EAD significantly outperforms contemporary advanced defense methods in both effectiveness and generalization based on two safety-critical tasks, including face recognition and object detection.

## 2 BACKGROUND

In this section, we introduce the threat from adversarial patches under a 3D environment and their corresponding defense strategy. Given a scene $\mathbf{x} \in \mathcal{X}$ with its ground-truth label $y \in \mathcal{Y}$, the perception model $f : \mathcal{O} \to \mathcal{Y}$ aims to predict the scene annotation $y$ using the image observation $o_i \in \mathcal{O}$. The observation $o_i$ is derived from scene $\mathbf{x}$ conditioned on the camera's state $s_i$ (*e.g.*, camera's position and viewpoint). The function $\mathcal{L}(\cdot)$ denotes a task-specific loss function (*e.g.*, cross-entropy loss for classification).

**Adversarial patches.** Though adversarial patches are designed to manipulate a specific region of an image to mislead the image classifiers (Brown et al., 2017), they have now evolved to deceive various perception models (Sharif et al., 2016; Song et al., 2018) under 3D environment (Eykholt et al., 2018; Yang et al., 2022; Zhu et al., 2023; Yang et al., 2023). Formally, an adversarial patch $p$ is

introduced to the image observation $o_i$, and results in erroneous prediction of the perception model $f$. Typically, the generation of adversarial patches in 3D scenes (Zhu et al., 2023) emphasizes solving the optimization problem presented as:

$$\max_p \mathbb{E}_{s_i} \mathcal{L}(f(A(o_i, p; s_i)), y), \tag{1}$$

where $A(\cdot)$ denotes the applying function to project adversarial patch $p$ from the 3D physical space to 2D perspective views of camera observations $o_i$ based on the camera's state $s_i$. This approach provides a more general 3D formulation for adversarial patches. By specifying the state $s_i$ as a mixture of 2D transformations and patch locations, it aligns with the 2D formulation by Brown et al. (2017). To ensure clarity in subsequent sections, we define the set of adversarial patches $\mathbb{P}_\mathbf{x}$ that can deceive the model $f$ under scene $\mathbf{x}$ as:

$$\mathbb{P}_\mathbf{x} = \{p \in [0, 1]^{H_p \times W_p \times C} : \mathbb{E}_{s_i} f(A(o_i, p; s_i)) \neq y\}, \tag{2}$$

where $H_p$ and $W_p$ denote the patch's height and width. In practice, the approximate solution of $\mathbb{P}_\mathbf{x}$ is determined by applying specific methods (Goodfellow et al., 2014; Madry et al., 2017; Carlini & Wagner, 2017; Dong et al., 2018) to solve the problem in Eq. (1).

**Adversarial defenses against patches.** A multitude of defensive strategies, including both empirical (Dziugaite et al., 2016; Hayes, 2018; Naseer et al., 2019; Rao et al., 2020; Xu et al., 2023) and certified defenses (Li et al., 2018; Zhang & Wang, 2019; Xiang et al., 2021), have been suggested to safeguard perception models against patch attacks. However, the majority of contemporary defense mechanisms fall under the category of **passive defenses**, as they primarily rely on information obtained from passively-received monocular observation and prior adversary's knowledge to alleviate adversarial effects. In particular, adversarial training approaches (Madry et al., 2017; Wu et al., 2019; Rao et al., 2020) strive to learn a more robust perception model $f$, while adversarial purification-based techniques (Hayes, 2018; Naseer et al., 2019; Liu et al., 2022; Xu et al., 2023) introduce an auxiliary purifier $g : \mathcal{O} \rightarrow \mathcal{O}$ to "detect and remove" (Liu et al., 2022) adversarial patches in image observations, subsequently enhancing robustness through the amended perception model $f \circ g$.

**Embodied perception.** In the realm of embodied perception (Aloimonos et al., 1988; Bajcsy, 1988), an embodied agent can navigate its environment to optimize perception or enhance the efficacy of task performance. Such concept has found applications in diverse tasks such as object detection (Yang et al., 2019; Chaplot et al., 2021; Kotar & Mottaghi, 2022; Jing & Kong, 2023), 3D pose estimation (Doumanoglou et al., 2016; Ci et al., 2023) and 3D scene understanding (Das et al., 2018; Ma et al., 2022). To our knowledge, our work is the first to integrate an embodied active strategy to diminish the high uncertainty derived from adversarial patches to enhance model robustness.

## 3 METHODOLOGY

We first introduce Embodied Active Defense (EAD) that utilizes embodied recurrent feedback to counter adversarial patches in Sec. 3.1. We then provide a theoretical analysis of the defensive ability of EAD in Sec. 3.2. Lastly, the technical implementation details of EAD are discussed in Sec. 3.3.

### 3.1 EMBODIED ACTIVE DEFENSE

Conventional passive defenses usually process single observation $o_i$ for countering adversarial patches, thereby neglecting the rich scenic information available from alternative observations acquired through proactive movement (Ronneberger et al., 2015; He et al., 2017). In this paper, we propose the **Embodied Active Defense (EAD)**, which emphasizes active engagement within a scene and iteratively utilizes environmental feedback to improve the robustness of perception against adversarial patches. Unlike previous methods, EAD captures a series of observations through strategic actions, instead of solely relying on a single passively-received observation.

EAD is comprised of two recurrent models that emulate the cerebral structure underlying active human vision, each with distinct functions. The **perception model** $f(\cdot; \boldsymbol{\theta})$, parameterized by $\boldsymbol{\theta}$, is dedicated to visual perception by fully utilizing the contextual information within temporal observations from the external world. It leverages the observation $o_t$ and the prevailing internal belief $b_{t-1}$ on the scene to construct a better representation of the surrounding environment $b_t$ in a recurrent paradigm and simultaneously predict scene annotation, as expressed by:

$$\{\overline{y}_t, b_t\} = f(o_t, b_{t-1}; \boldsymbol{\theta}). \tag{3}$$

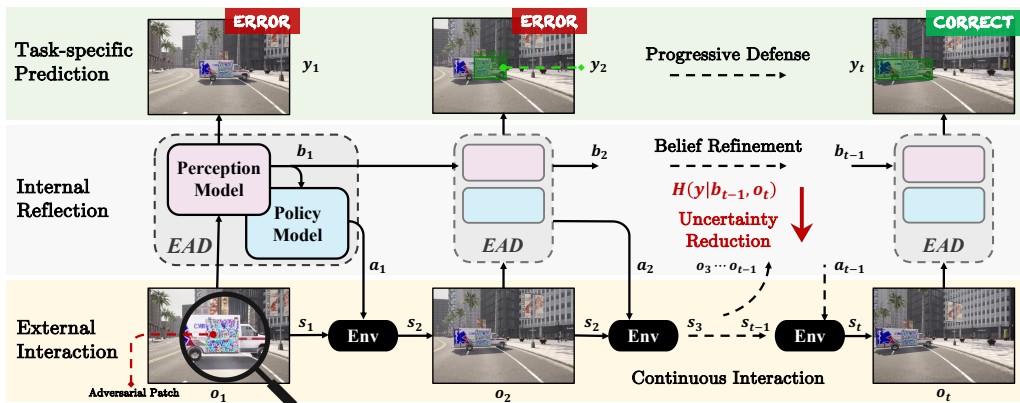

Figure 1: An overview of our proposed embodied activate defense. The perception model utilizes observation $o_t$ from the external world and previous internal belief $b_{t-1}$ on the scene to refine a representation of the surrounding environment $b_t$ and simultaneously make task-specific prediction $y_t$. The policy model generates strategic action $a_t$ in response to shared environmental understanding $b_t$. As the perception process unfolds, the initially high informative uncertainty $H(y|b_{t-1}, o_t)$ caused by adversarial patch monotonically decreases.

The subsequent **policy model** $\pi(\cdot; \boldsymbol{\phi})$, parameterized by $\boldsymbol{\phi}$, governs the visual control of movement. Formally, It derives $a_t$ rooted in the collective environmental understanding $b_t$ sustained by the perception model, as formalized by $a_t \sim \pi(b_t; \boldsymbol{\phi})$.

**Leveraging recurrent feedback from the environment.** Formally, the EAD's interaction with its environment (as depicted in Figure 1) follows a simplified Partially-Observable Markov Decision Process (POMDP) framework, facilitating the proactive exploration of EAD within the scene $\mathbf{x}$. The interaction process is denoted by $\mathcal{M}(\mathbf{x}) := \langle \mathcal{S}, \mathcal{A}, \mathcal{T}, \mathcal{O}, \mathcal{Z} \rangle$. Here, each $\mathbf{x}$ determines a specific MDP by parameterizing a transition distribution $\mathcal{T}(s_{t+1}|s_t, a_t, \mathbf{x})$ and an observation distribution $\mathcal{Z}(o_t|s_t, \mathbf{x})$. At every moment $t$, EAD obtains an observation $o_t \sim \mathcal{Z}(\cdot|s_t, \mathbf{x})$ based on current state $s_t$. It utilizes $o_t$ as an environmental feedback to refine the understanding of environment $b_t$ through perception model $f(\cdot; \boldsymbol{\theta})$. Such a recurrent perception mechanism is pivotal for the stability of human vision (Thomas, 1999; Kok et al., 2012; Kar et al., 2019). Subsequently, the EAD model executes actions $a_t \sim \pi(b_t; \boldsymbol{\phi})$, rather than remaining static and passively assimilating observation. Thanks to the policy model, EAD is capable of determining the optimal action, ensuring the acquisition of the most informative feedback from the scene, thereby enhancing perceptual efficacy.

**Training EAD against adversarial patches.** Within the intricately constructed EAD model, we introduce a specialized learning algorithm tailored for EAD to counter adversarial patches. Considering a data distribution $\mathcal{D}$ with paired data $(\mathbf{x}, y)$ and adversarial patches $p \in \mathbb{P}_{\mathbf{x}}$ crafted with *unknown attack techniques*, our proposed approach optimizes parameters $\boldsymbol{\theta}$ and $\boldsymbol{\phi}$ by minimizing the expected loss amid threats from adversarial patches, where the adversarial patch $p$ contaminates the observation $o_t$ and yields $o'_t = A(o_t, p; s_t)$. Consequently, the learning of EAD to mitigate adversarial patches is cast as an optimization problem:

$$\min_{\boldsymbol{\theta}, \boldsymbol{\phi}} \mathbb{E}_{(\mathbf{x}, y) \sim \mathcal{D}} \Big[ \sum_{p \in \mathbb{P}_{\mathbf{x}}} \mathcal{L}(\overline{y}_t, y) \Big], \text{with } \{\overline{y}_t, b_t\} = f(A(o_t, p; s_t), b_{t-1}; \boldsymbol{\theta}), \ a_t \sim \pi(\cdot|b_t; \boldsymbol{\phi})$$

$$\text{s.t.} \quad o_t \sim \mathcal{Z}(\cdot|s_t, \mathbf{x}), \quad s_t \sim \mathcal{T}(\cdot|s_{t-1}, a_{t-1}, \mathbf{x}), \tag{4}$$

where $t = 1, \ldots, \tau$, $\overline{y}_t$ denotes the model's prediction at timestep $t$ and $\tau$ represents the maximum horizon length of EAD. Importantly, the loss function $\mathcal{L}$ remains agnostic to tasks, underscoring the remarkable versatility of EAD. This flexibility ensures EAD to offer robust defenses across diverse perception tasks. We further delve into the efficacy of EAD through an information-theoretic lens.

### 3.2 A PERSPECTIVE FROM INFORMATION THEORY

Drawing inspiration from active human vision, we introduce EAD in Sec. 3.1 as a strategy to enhance a model's robustness against adversarial patches. However, due to the black-box nature of neural networks, the fundamental driver behind this defensive tactic remains elusive. To delve deeper into model behaviors, we examine a generic instance of EAD in Eq. (4), where the agent employs the

InfoNCE objective (Oord et al., 2018). This is succinctly reformulated[1] as:

$$\max_{\boldsymbol{\theta},\boldsymbol{\phi}} \ \mathbb{E}_{(\mathbf{x}^{(j)},y^{(j)})\sim\mathcal{D}} \left[ -\sum_{j=1}^{K} \log \frac{e^{-S(f(b_{t-1}^{(j)},o_t^{(j)};\boldsymbol{\theta}),y^{(j)})}}{\sum_{\hat{y}^{(j)}} e^{-S(f(b_{t-1}^{(j)},o_t^{(j)};\boldsymbol{\theta}),\hat{y}^{(j)})}} \right], \tag{5}$$

where $S(\cdot) : \mathcal{Y} \times \mathcal{Y} \to \mathbb{R}$ measures the similarity between predicted scene annotation and ground truth, and $K$ denotes the size of data batch $\{(\mathbf{x}^{(j)}, y^{(j)})\}_{j=1}^{K}$ sampled from $\mathcal{D}$. Under the realm of embodied perception, the target task with InfoNCE objective is to match the observations collected from given scene $\mathbf{x}$ with its annotation $y$ other modalities like CLIP (Radford et al., 2021), where the annotations are usually text for tasks like captioning given scene (Jin et al., 2015) or answering the question about scene (Ma et al., 2022).

**Theorem 3.1** (Proof in Appendix A.1). *For mutual information between current observation $o_t$ and scene annotation $y$ conditioned on previous belief $b_{t-1}$, denoted as $I(o_t; y|b_{t-1})$, we have:*

$$I(o_t; y|b_{t-1})) \geq \mathbb{E}_{(\mathbf{x}^{(j)},y^{(j)})\sim\mathcal{D}} \left[ \sum_{j=1}^{K} \log \frac{q_{\boldsymbol{\theta}}(y^{(j)}|b_{t-1}^{(j)}, o_t^{(j)})}{\sum_{\hat{y}^{(j)}} q_{\boldsymbol{\theta}}(\hat{y}^{(j)}|b_{t-1}^{(j)}, o_t^{(j)})} \right] + \log(K), \tag{6}$$

*where $q_{\boldsymbol{\theta}}(y|o_1, \cdots, o_t)$ denotes variational distribution for conditional distribution $p(y|o_1, \cdots, o_t)$ with samples $\{(\mathbf{x}^{(j)}, y^{(j)})\}_{j=1}^{K}$.*

**Remark.** To bridge the lower bound of conditional mutual information with the objective, we can rewrite $q_{\boldsymbol{\theta}}(y|b_{t-1}, o_t)$ with the similarity term in Eq. (4) served as score function, and obtain $q_{\theta}(y|b_{t-1}, o_t) := p(b_{t-1}, o_t)e^{-S(f(b_{t-1}, o_t; \boldsymbol{\theta})|y)}$. Then, this term is equivalent to the negative InfoNCE objective for EAD in Eq. (4). It indicates that the training procedure for EAD is actually to indirectly maximize the conditional mutual information, thereby leading the agent to learn an action policy aiming to collect an observation $o_t$ to better determine the task-designated annotation $y$. And the bound will be tighter as the batch size $K$ increases.

**Definition 3.1** (Greedy Informative Exploration). *A greedy informative exploration, denoted by $\pi^*$, is an action policy which, at any timestep $t$, chooses an action $a_t$ that maximizes the decrease in the conditional entropy of a random variable $y$ given a new observation $o_t$ resulting from taking action $a_t$. Formally,*

$$\pi^* = \arg\max_{\pi\in\Pi} [H(y|b_{t-1}) - H(y|b_{t-1}, o_t)], \tag{7}$$

*where $H(\cdot)$ denotes the entropy, $\Pi$ is the space of all policies.*

**Remark.** The conditional entropy $H(y|b_{t-1})$ quantifies the uncertainty of the target $y$ given previously sustained belief $b_{t-1}$, while $H(y|b_{t-1}, o_t)$ denotes the uncertainty of $y$ with extra observation $o_t$. Although not optimal throughout the whole trajectory, the **greedy informative exploration** serves as a relatively efficient baseline for agents to rapidly understand their environment by performing continuous actions and observations. The efficiency of the greedy policy is empirically demonstrated in Appendix D.7.

Given unlimited model capacity and data samples, the optimal policy model $\pi_{\phi}^*$ in problem (5) is a greedy informative exploration policy. The optimization procedure in Eq. (5) simultaneously estimates the mutual information and improves the action policy with its guidance. Owing to the mutual information being equivalent to the conditional entropy decrease (detailed proof in Appendix A.2), it leads the action policy approaching greedy informative exploration.

Therefore, we theoretically demonstrate the effectiveness of EAD from the perspective of information theory. A well-learned EAD model for contrastive task adopts a **greedy informative policy** to explore the environment, utilizing the rich context information to reduce the abnormally high uncertainty (Smith & Gal, 2018; Deng et al., 2021) of scenes caused by adversarial patches.

## 3.3 IMPLEMENTATION TECHNIQUES

The crux of EAD's learning revolves around the solution of the optimization problem (4). However, due to the intractability of the environment dynamics, namely the observation $\mathcal{Z}$ and transition $\mathcal{T}$, it is not feasible to directly optimize the policy model parameters using gradient descent. Concurrently,

---

[1]Constraints in Eq. (4) and belief $b_t$ in the output have been excluded for simplicity.

---

**Algorithm 1** Learning Embodied Active Defense

---

**Require:** Training data $\mathcal{D}$, number of epochs $N$, loss function $\mathcal{L}$, perception model $f(\cdot; \boldsymbol{\theta})$, policy model $\pi(\cdot; \boldsymbol{\phi})$, differentiable observation function $Z$ and transition function $T$, max horizon length $\tau$, uniform superset approximation for adversarial patches $\tilde{\mathbb{P}}$ and applying function $A$.
**Ensure:** The parameters $\boldsymbol{\theta}, \boldsymbol{\phi}$ of the learned EAD model.
 1: **for** epoch $\leftarrow 1$ **to** $N$ **do**
 2:     $\boldsymbol{X}, \boldsymbol{Y} \leftarrow$ sample a batch from $\mathcal{D}_{\text{train}}$;
 3:     $\boldsymbol{P} \leftarrow$ sample a batch from from $\tilde{\mathbb{P}}$;
 4:     $t \leftarrow 1, \boldsymbol{A}_0 \leftarrow \texttt{null}, \boldsymbol{B}_0 \leftarrow \texttt{null}$, randomly initialize world state $\boldsymbol{S}_0$;
 5:     **repeat**
 6:         $\boldsymbol{S}_t \leftarrow T(\boldsymbol{S}_{t-1}, \boldsymbol{A}_{t-1}, \boldsymbol{X}), \boldsymbol{O}_t \leftarrow Z(\boldsymbol{S}_t, \boldsymbol{X}), \boldsymbol{O}'_t \leftarrow A(\boldsymbol{O}_t, \boldsymbol{P}; \boldsymbol{S}_t)$;     ▷ *Compute **observations***
 7:         $\boldsymbol{B}_t, \overline{\boldsymbol{Y}}_t \leftarrow f(\boldsymbol{B}_{t-1}, \boldsymbol{O}'_t; \boldsymbol{\theta}), \boldsymbol{A}_t \leftarrow \pi(\boldsymbol{B}_t; \boldsymbol{\phi})$;         ▷ *Compute **beliefs**, **actions** and **predictions***
 8:         $\boldsymbol{S}_{t+1} \leftarrow T(\boldsymbol{S}_t, \boldsymbol{A}_t, \boldsymbol{X}), \boldsymbol{O}_{t+1} \leftarrow Z(\boldsymbol{S}_{t+1}, \boldsymbol{X}), \boldsymbol{O}'_{t+1} \leftarrow A(\boldsymbol{O}_{t+1}, \boldsymbol{P}; \boldsymbol{S}_{t+1})$;
 9:         $\boldsymbol{B}_{t+1}, \overline{\boldsymbol{Y}}_{t+1} \leftarrow f(\boldsymbol{B}_t, \boldsymbol{O}'_{t+1}; \boldsymbol{\theta}), \boldsymbol{A}_{t+1} \leftarrow \pi(\boldsymbol{B}_{t+1}; \boldsymbol{\phi})$;
10:         $\mathcal{L}_{t+1} \leftarrow \mathcal{L}(\overline{\boldsymbol{Y}}_{t+1}, \boldsymbol{Y})$;
11:         Update $\boldsymbol{\theta}, \boldsymbol{\phi}$ using $\nabla \mathcal{L}_{t+1}$;
12:         $t \leftarrow t + 2$;
13:     **until** $t > \tau$
14: **end for**

---

the scene-specific adversarial patches $\mathbb{P}_x$ cannot be derived analytically. To address these challenges, we propose two approximation methods to achieve near-accurate solutions. Empirical evidence from experiments demonstrates that these approaches are effective in reliably solving problem (4).

**Deterministic and differentiable approximation for environments.** Formally, we employ the Delta distribution to deterministically model the transition $\mathcal{T}$ and observation $\mathcal{Z}$. For instance, the approximation of $\mathcal{T}$ is expressed as $\mathcal{T}(s_{t+1}|s_t, a_t, \mathbf{x}) = \delta(s_{t+1} - T(s_t, a_t, \mathbf{x}))$, where $T : \mathcal{S} \times \mathcal{A} \times \mathcal{X} \rightarrow \mathcal{S}$ denotes the mapping of the current state and action to the most probable next state, and $\delta(\cdot)$ represents the Delta distribution. Additionally, we use advancements in differentiable rendering (Kato et al., 2020) to model deterministic observations by rendering multi-view image observations differentiably, conditioned on camera parameters deduced from the agent's current state.

These approximations allow us to create a connected computational graph between the policy and perception models (Abadi, 2016; Paszke et al., 2019), thereby supporting the use of backpropagation (Werbos, 1990; Williams & Peng, 1990) to optimize the policy model's parameter $\phi$ via supervised learning. To maximize supervision signal frequency and minimize computational overhead, we update model parameters $\boldsymbol{\theta}$ and $\boldsymbol{\phi}$ every second step, right upon the formation of the minimal computational graph that includes the policy model. This approach allows us to seamlessly integrate the passive perception models into EAD to enhance resistance to adversarial patches.

**Uniform superset approximation for adversarial patches.** The computation of $\mathbb{P}_x$ typically necessitates the resolution of the inner maximization in Eq. (4). However, this is not only computationally expensive (Wong et al., 2020) but also problematic as inadequate assumptions for characterizing adversaries can hinder the models' capacity to generalize across diverse, unseen attacks (Laidlaw et al., 2020). To circumvent these limitations, we adopt an assumption-free strategy that solely relies on uniformly sampled patches, which act as surrogate representatives encompassing a broad spectrum of potential adversarial examples. Formally, the surrogate set of adversarial patches is defined as:

$$\tilde{\mathbb{P}} \coloneqq \{p_i\}_{i=1}^N, \quad p_i \sim \mathcal{U}(0,1)^{H_p \times W_p \times C}, \tag{8}$$

where $N$ represents the size of the surrogate patch set $\tilde{\mathbb{P}}$. In the context of implementation, $N$ corresponds to the training epochs, suggesting that as $N \rightarrow \infty$, we achieve $\mathbb{P}_\mathbf{x} \subseteq \tilde{\mathbb{P}}$. By demanding the EAD to address various patches, the active perception significantly bolsters its resilience against adversarial patches (refer to Sec. 4.1). The overall training procedure is outlined in Algorithm 1.

## 4 EXPERIMENTS

In this section, we first introduce the experimental environment, and then present extensive experiments to demonstrate the effectiveness of EAD on face recognition (FR) and object detection.

**Experimental Environment**. To enable EAD's free navigation and observation collection, a manipulable simulation environment is indispensable for both training and testing. For alignment with

Table 1: Standard accuracy and adversarial attack success rates on FR models. [†] denotes the methods involved with adversarial training. Columns with *Adpt* represent results under *adaptive attack*, and the adaptive attack against EAD optimizes the patch with an expected gradient over the distribution of possible action policy.

| Method | Acc. (%) | Dodging ASR (%) | | | | | Impersonation ASR (%) | | | | |
|---|---|---|---|---|---|---|---|---|---|---|---|
| | | MIM | EoT | GenAP | 3DAdv | Adpt | MIM | EoT | GenAP | 3DAdv | Adpt |
| Undefended | 88.86 | 100.0 | 100.0 | 99.0 | 98.0 | 100.0 | 100.0 | 100.0 | 99.0 | 89.0 | 100.0 |
| JPEG | 89.98 | 98.0 | 99.0 | 95.0 | 88.0 | 100.0 | 99.0 | 100.0 | 99.0 | 93.0 | 99.0 |
| LGS | 83.50 | 49.5 | 52.6 | 74.0 | 77.9 | 78.9 | 5.1 | 7.2 | 33.7 | 30.6 | 38.4 |
| SAC | 86.83 | 73.5 | 73.2 | 92.8 | 78.6 | 65.2 | 6.1 | 9.1 | 67.7 | 64.6 | 48.0 |
| PZ | 87.58 | 6.9 | 8.0 | 58.4 | 57.1 | 88.9 | 4.1 | 5.2 | 59.4 | 45.8 | 89.8 |
| SAC[†] | 80.55 | 78.8 | 78.6 | 79.6 | 85.8 | 85.0 | 3.2 | 3.2 | 18.9 | 22.1 | 51.7 |
| PZ[†] | 85.85 | 6.1 | 6.2 | 14.3 | 20.4 | 69.4 | **3.1** | 3.2 | 19.1 | 27.4 | 61.0 |
| DOA[†] | 79.55 | 75.3 | 67.4 | 87.6 | 95.5 | 95.5 | 95.5 | 89.9 | 96.6 | 89.9 | 89.9 |
| **EAD (ours)** | **90.45** | **0.0** | **0.0** | **2.1** | **13.7** | **22.1** | 4.1 | **3.1** | **5.1** | **7.2** | **8.3** |

given vision tasks, we define the state as a composite of the camera's yaw and pitch and the action as the camera's rotation. This sets the transition function, so the core of the simulation environment revolves around the observation function, which renders a 2D image given the camera state. **1) Training environment.** As discussed in Sec. 3.3, model training demands differentiable environmental dynamics. To this end, we employ the cutting-edge 3D generative model, EG3D (Chan et al., 2022) for realistic differentiable rendering (see Appendix D.1 for details on simulation fidelity). **2) Testing environment.** In addition to evaluations conducted in EG3D-based simulations, we extend our testing for object detection within CARLA (Dosovitskiy et al., 2017), aiming to assess EAD in more safety-critical autonomous driving scenarios. The details can be found in Appendix C.

## 4.1 Evaluation on Face Recognition

**Evaluation setting.** We conduct our experiments on CelebA-3D, which we utilize GAN inversion (Zhu et al., 2016) with EG3D (Chan et al., 2022) to reconstruct 2D face image from CelebA into a 3D form. For standard accuracy, we sample $2,000$ test pairs from the CelebA and follow the standard protocol from LFW (Huang et al., 2007). As for robustness evaluation, we report the white-box attack success rate (ASR) on 100 identity pairs in both impersonation and dodging attacks (Yang & Zhu, 2023) with various attack methods, including MIM (Dong et al., 2018), EoT (Athalye et al., 2018b), GenAP (Xiao et al., 2021) and Face3Dadv (3DAdv) (Yang et al., 2022). Note that 3DAdv utilizes expectation over 3D transformations in optimization, rendering it robust to 3D viewpoint variation (within $\pm 15°$, refer to Appendix D.7). More details are described in Appendix D.1 & D.2.

**Implementation details.** For the visual backbone, we employ the pretrained IResNet-50 Arc-Face(Duta et al., 2021) with frozen weight in subsequent training. To implement the recurrent perception and policy, we adopt a variant of the Decision Transformer (Chen et al., 2021) to model the temporal process which uses feature sequences extracted by the visual backbone to predict a normalized embedding for FR. To expedite EAD's learning of efficient policies requiring minimal perceptual steps, we configure the max horizon length $\tau = 4$. Discussion about this horizon length is provided in Appendix D.7 and the details for EAD are elaborated in Appendix D.5.

**Defense baselines.** We benchmark EAD against a range of defense methods, including adversarial training-based Defense against Occlusion Attacks (DOA) (Wu et al., 2019), and purification-based methods like JPEG compression (JPEG) (Dziugaite et al., 2016), local gradient smoothing (LGS) (Naseer et al., 2019), segment and complete (SAC) (Liu et al., 2022) and PatchZero (PZ) (Xu et al., 2023). For DOA, we use rectangle PGD patch attacks (Madry et al., 2017) with 10 iterations and step size $2/255$. SAC and PZ require a patch segmenter to locate the area of the adversarial patch. Therefore, we train the segmenter with patches of Gaussian noise to ensure the same adversarial-agnostic setting as EAD. Besides, enhanced versions of SAC and PZ involve training with EoT-generated adversarial patches, denoted as SAC[†] and PZ[†]. More details are in Appendix D.4.

**Effectiveness of EAD.** Table 1 demonstrates both standard accuracy and robust performance against diverse attacks under a white-box setting. The patch is 8% of the image size. Remarkably, our approach outperforms previous state-of-the-art techniques that are agnostic to adversarial examples in both clean accuracy and defense efficacy. For instance, EAD reduces the attack success rate of 3DAdv by $84.3\%$ in impersonation and by $81.8\%$ in dodging scenarios. Our EAD even surpasses

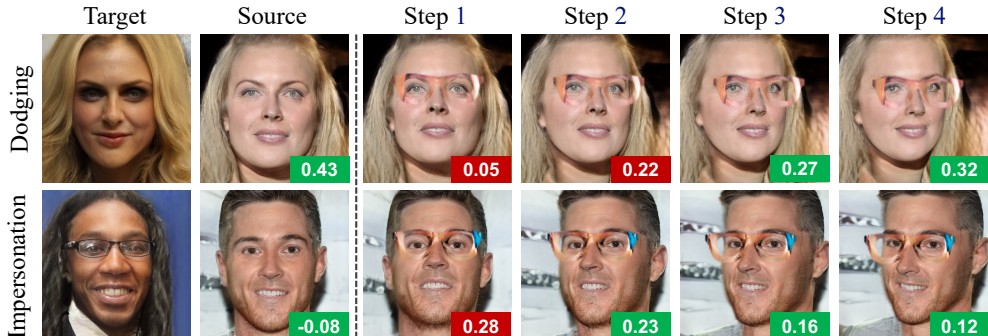

Figure 2: Qualitative results of EAD. The first two columns present the original image pairs, and the subsequent columns depict the interactive inference steps that the model took. The adversarial glasses are generated with 3DAdv, which are robust to 3D viewpoint variation. The computed optimal threshold is $0.24$ from $[-1, 1]$.

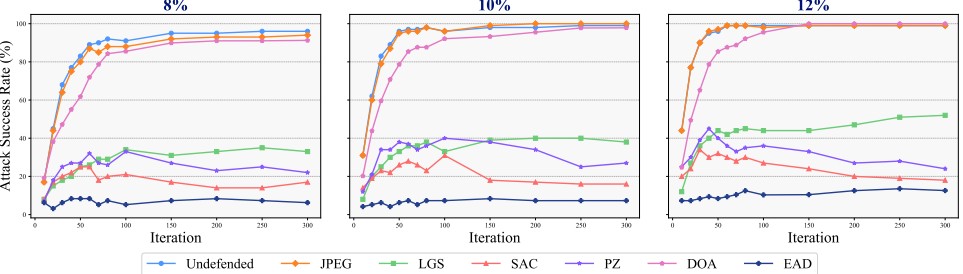

Figure 3: Comparative evaluation of defense methods across varying attack iterations with different adversarial patch sizes. The adversarial patches are crafted by 3DAdv for impersonation.

the undefended passive model in standard accuracy, which facilitates the reconciliation between robustness and accuracy (Su et al., 2018) through embodied perception. Furthermore, our method even outstrips baselines by incorporating adversarial examples during training. Although SAC[†] and PZ[†] are trained using patches generated via EoT (Athalye et al., 2018b), we still obtain superior performance which stems from effective utilization of the environmental feedback in active defense.

Figure 2 illustrates the defense process executed by EAD. While EAD may fooled at first glance, its subsequent active interactions with the environment progressively increase the similarity between the positive pair and decrease the similarity between the negative pair. Consequently, EAD effectively mitigates adversarial effects through the proactive acquisition of additional observations.

**Effectiveness against adaptive attack.** While the deterministic and differential approximation could enable backpropagation through the entire inference trajectory of EAD, the computational cost is prohibitive due to rapid GPU memory consumption as trajectory length $\tau$ increases. To overcome this, we resort to an approach similar to USAP that approximates the true gradient with the expected gradient over a surrogate uniform superset policy distribution. This necessitates an optimized patch to handle various action policies. Our adaptive attack implementation builds upon 3DAdv (Yang et al., 2022) using 3D viewpoint variations. We can see that EAD maintains its robustness against the most potent attacks. It further shows that EAD's defensive capabilities arise from the synergistic integration of its policy and perception models, rather than learning a short-cut strategy to neutralize adversarial patches from specific viewpoints. More details are in Appendix D.3.

**Generalization of EAD.** As illustrated in Table 1, despite no knowledge of adversaries, ours demonstrates strong generalizability across various unseen adversarial attack methods. It is partially attributed to EAD's capability to dynamically interact with its environment, enabling it to adapt and respond to new types of attacks. Additionally, we assess the model's resilience across a wide range of patch sizes and attack iterations. Trained solely on patches constituting $10\%$ of the image, EAD maintains a notably low attack success rate even when the patch size and attack iteration increases, as shown in Figure 3. This resilience can be attributed to EAD's primary reliance on environmental information rather than patterns of presupposed adversaries, thus avoiding overfitting specific attack types. The details are available in Appendix D.7.

**Ablation study.** We conduct ablation studies within EAD in Table 2. **1) Effectiveness of recurrent feedback.** We initially demonstrate the role of recurrent feedback, *i.e.*, reflecting on prior belief

Table 2: Standard accuracy and white-box impersonation adversarial attack success rates on ablated models. For the model with random action policy, we report the mean and standard deviation under five rounds.

| Method | Acc. (%) | Attack Success Rate (%) | | | | |
|---|---|---|---|---|---|---|
| | | MIM | EoT | GenAP | 3DAdv | Adpt |
| Undefended | 88.86 | 100.0 | 100.0 | 99.00 | 98.00 | 98.00 |
| Random Movement | 90.38 | $4.17 \pm 2.28$ | $5.05 \pm 1.35$ | $8.33 \pm 2.21$ | $76.77 \pm 3.34$ | $76.77 \pm 3.34$ |
| Perception Model | 90.22 | $18.13 \pm 4.64$ | $18.62 \pm 2.24$ | $22.19 \pm 3.97$ | $30.77 \pm 1.81$ | $31.13 \pm 3.01$ |
| + Policy Model | 89.85 | **3.09** | 4.12 | 7.23 | 11.34 | 15.63 |
| + USAP | **90.45** | 4.12 | **3.07** | **5.15** | **7.21** | **8.33** |

| Ground Truth | Step 1 | Step 2 | Step 3 | Step 4 |
|---|---|---|---|---|

Figure 4: Qualitative results of EAD on object detection. The adversarial patches are generated using MIM and attached to the billboards within the scene, leading to the "disappearing" of the target vehicle. The setting is from the CARLA-GEAR. These images illustrate the model's interactive inference steps to counter the patches.

with a comprehensive fusion model, in achieving robust performance. EAD with only the perception model significantly surpasses both the undefended baseline and passive FR model with multi-view ensembles (Random Movement). Notably, the multi-view ensemble model fails to counteract 3DAdv, corroborating that EAD's defensive strength is not merely a function of the vulnerability of adversarial examples to viewpoint transformations. **2) Utility of learned policy.** We then investigate the benefits of the learned policy on robust perception by comparing Clean-Data EAD with the perception model adopting random action. The learned policy markedly enhances resistance to various attacks. The findings supporting the effectiveness and efficiency of the learned policy are furnished in Appendix D.7. **3) Influence of patched data.** By employing EAD to handle a diverse set of patches sampled from a surrogate uniform superset during training, the robustness of EAD is further augmented.

## 4.2 EVALUATION ON OBJECT DETECTION

**Experimental setup.** We further apply EAD to object detection to verify its adaptability. We train EAD with a simulation environment powered by EG3D (Chan et al., 2022), and evaluate the performance of EAD on robustness evaluation API provided by CARLA-GEAR (Nesti et al., 2022). For the object detector, we use the pretrained Mask-RCNN (He et al., 2017) on COCO (Lin et al., 2014). We assess model robustness on 360 test scenes featuring patches attached to billboards and re-

Table 3: The mAP (%) of Mask-RCNN under different adversarial attacks.

| Method | Clean | MIM | TIM | SIB |
|---|---|---|---|---|
| Undefended | **46.6** | 30.6 | 34.2 | 31.2 |
| LGS | 45.8 | 36.5 | 37.8 | 36.4 |
| SAC[†] | 46.5 | 33.3 | 35.1 | 32.5 |
| PZ[†] | 46.3 | 33.2 | 35.1 | 32.9 |
| **EAD (ours)** | **46.6** | **39.4** | **39.3** | **39.5** |

port the mean Average Precision (mAP). For attacks, we use methods including MIM (Dong et al., 2018), TIM (Dong et al., 2019) and SIB (Zhao et al., 2019) with iterations set at 150 and a step size of 0.5. Specifically, we augment MIM and TIM by incorporating EoT over 2D transformations to enhance patch resilience against viewpoint changes. Meanwhile, SIB demonstrates robustness with varying distances ($1 \sim 25$ m) and angles ($\pm 60°$). (Zhao et al., 2019). The patch size varies with the billboard size, constituting $4 \sim 8\%$ of the image size. Further details can be found in Appendix E.

**Experimental results.** As evidenced in Table 3, our approach surpasses the others in both clean and robust performances. Furthermore, EAD retains its robustness and effectively generalizes against a variety of unencountered attacks, thus substantiating our previous assertions in FR. A visualization of the defense process is provided in Figure 4. More detailed results are provided in Appendix E.5.

## 5 CONCLUSION

In this paper, we introduce a novel proactive strategy against adversarial patches, named EAD. Inspired by active human vision, EAD merges external visual signals and internal cognitive feedback via two recurrent sub-modules. It facilitates the derivation of proactive strategic actions and continuous refinement of target understanding by recurrently leveraging environmental feedback. Experiments validate the effectiveness and generalizability of EAD in enhancing defensive capabilities.

## ETHICS STATEMENT

Embodied Active Defense (EAD) shows promise as protection against adversarial patch attacks under 3D physical environment. In the field of adversarial defense, EAD presents a promising direction for enhancing robustness. However, EAD could also impact the performance of some applications like face watermarking and facial privacy protection, which rely on generating adversarial perturbations. We advocate for responsible use, particularly in contexts where bias could occur. Although EAD effectively counters adversarial attacks, it may require additional hardware or computational resources to enable dynamic interactions and process temporal information. We report no conflicts of interest and adhere to the highest standards of research integrity.

## ACKNOWLEDGEMENTS

This work was supported by the National Natural Science Foundation of China (Nos. 62276149, U2341228, 62061136001, 62076147), BNRist (BNR2022RC01006), Tsinghua Institute for Guo Qiang, Tsinghua-OPPO Joint Research Center for Future Terminal Technology, and the High Performance Computing Center, Tsinghua University. Y. Dong was also supported by the China National Postdoctoral Program for Innovative Talents and Shuimu Tsinghua Scholar Program. J. Zhu was also supported by the XPlorer Prize.

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

# A    PROOFS AND ADDITIONAL THEORY

## A.1    PROOF OF THEOREM 3.1

*Proof.* For series of observations $\{o_1, \cdots o_t\}$ and previously maintained belief belief $b_{t-1}$ are determined by the scene $\mathbf{x}$, we expand the left-hand side of Eq. (6) as follows:

$$I(o_t; y|b_{t-1})$$
$$= \mathbb{E}_{\mathbf{x},y} \log \frac{p(b_{t-1})p(o_1, \cdots o_t, y)}{p(b_{t-1}, o_{t-1}, y)p(b_{t-1}, o_t)}$$
$$= \mathbb{E}_{\mathbf{x},y} \log \frac{p(y|b_{t-1}, o_t)}{p(y|b_{t-1})}. \tag{9}$$

By multiplying and dividing the integrand in Eq. (9) by variational distribution $q_\theta(y|o_1, \cdots o_t)$, we have:

$$\mathbb{E}_{\mathbf{x},y} \log \frac{p(y|b_{t-1}, o_t)q_\theta(y|b_{t-1}o_t)}{p(y|b_{t-1})q_\theta(y|b_{t-1}, o_t)}$$
$$= \mathbb{E}_{\mathbf{x},y}[\log \frac{q_\theta(y|b_{t-1}, o_t)}{p(y|b_{t-1})} - D_{\mathrm{KL}}(p(y|b_{t-1}, o_t)\|q_\theta(y|b_{t-1}, o_t))].$$

Due to the non-negativity of KL-divergence, we have a lower bound for mutual information:

$$\mathbb{E}_{\mathbf{x},y}[\log \frac{q_\theta(y|b_{t-1}, o_t)}{p(y|b_{t-1})} - D_{\mathrm{KL}}(p(y|b_{t-1}, o_t)\|q_\theta(y|b_{t-1}, o_t))]$$
$$\geq \mathbb{E}_{\mathbf{x},y} \log \frac{q_\theta(y|b_{t-1}, o_t)}{p(y|b_{t-1}, o_{t-1})}$$
$$= \mathbb{E}_{\mathbf{x},y} \log q_\theta(y|b_{t-1}, o_t) + H(y|b_{t-1}), \tag{10}$$

where $H(y|b_{t-1})$ denotes the conditional entropy of $y$ given belief $b_{t-1}$ and Eq. (10) is well known as Barber and Agakov bound (Barber & Agakov, 2004). Then, we choose an energy-based variational family that uses a *critic* $\mathcal{E}_\theta(b_{t-1}, o_t, y)$ and is scaled by the data density $p(b_{t-1}, o_t)$:

$$q_\theta(y|b_{t-1}, o_t) = \frac{p(y|b_{t-1})}{Z(b_{t-1}, o_t)} e^{\mathcal{E}_\theta(b_{t-1}, o_t, y)}, \tag{11}$$

where $Z(b_{t-1}, o_t) = \mathbb{E}_y e^{\mathcal{E}_\theta(b_{t-1}, o_t, y)}$. By substituting Eq. (11) distribution into Eq. (10), we have:

$$\mathbb{E}_{\mathbf{x},y} \log q_\theta(y|b_{t-1}, o_t) + H(y|b_{t-1})$$
$$= \mathbb{E}_{\mathbf{x},y}[\mathcal{E}_\theta(b_{t-1}, o_t, y)] - \mathbb{E}_{\mathbf{x}}[\log Z(b_{t-1}, o_t)], \tag{12}$$

which is the unnormalized version of the Barber and Agakov bound. By applying inequality $\log Z(b_{t-1}, o_t) \leq \frac{Z(b_{t-1}, o_t)}{a(b_{t-1}, o_t)} + \log[a(b_{t-1}, o_t)] - 1$ for any $a(b_{t-1}, o_t) > 0$, and the bound is tight when $a(b_{t-1}, o_t) = Z(b_{t-1}, o_t)$. Therefore, we have a tractable upper bound, which is known as a tractable unnormalized version of the Barber and Agakov lower bound on mutual information:

$$\mathbb{E}_{\mathbf{x},y}[\mathcal{E}_\theta(b_{t-1}, o_t, y)] - \mathbb{E}_{\mathbf{x}}[\log Z(b_{t-1}, o_t)]$$
$$\geq \mathbb{E}_{\mathbf{x},y}[\mathcal{E}_\theta(b_{t-1}, o_t, y)] - \mathbb{E}_{\mathbf{x}}[\frac{\mathbb{E}_y[e^{\mathcal{E}_\theta(b_{t-1}, o_t, y)}]}{a(b_{t-1}, o_t)} + \log[a(b_{t-1}, o_t)] - 1]$$
$$= 1 + \mathbb{E}_{\mathbf{x},y}[\log \frac{e^{\mathcal{E}_\theta(b_{t-1}, o_t, y)}}{a(b_{t-1}, o_t)}] - \mathbb{E}_{\mathbf{x}}[\frac{\mathbb{E}_y e^{\mathcal{E}_\theta(b_{t-1}, o_t, y)}}{a(b_{t-1}, o_t)}]. \tag{13}$$

To reduce variance, we leverage multiple samples $\{x^{(j)}, y^{(j)}\}_{j=1}^K$ from $\mathcal{D}$ to implement a low-variance but high-bias estimation for mutual information. For other observation trajectory which is

originated from other scene $(x^{(j)}, y^{(j)})$ $(j \neq i)$, we have its annotations $\{y^{(j)}\}_{j=1, j \neq i}^{K}$ independent from $x^{(i)}$ and $y^{(i)}$, we have:

$$a(b_{t-1}, o_t) = a(b_{t-1}, o_t; y^{(1)}, \cdots, y^{(K)}).$$

Then, we are capable of utilize additional samples$\{x^{(j)}, y^{(j)}\}_{j=1}^{K}$ to build a Monte-Carlo estimate of the function $Z(b_{t-1}, o_t)$:

$$a(b_{t-1}, o_t; y_1, \cdots y_K) = m(b_{t-1}, o_t; y^{(1)}, \cdots y^{(K)}) = \frac{1}{K} \sum_{j=1}^{K} e^{\mathcal{E}_\theta(y^{(j)}|b_{t-1}, o_t)}.$$

To estimate the bound over $K$ samples, the last term in Eq. (13) becomes constant 1:

$$\mathbb{E}_{\mathbf{x}}\Big[\frac{\mathbb{E}_{y^{(1)}, \cdots, y^{(K)}} e^{\mathcal{E}_\theta(b_{t-1}, o_t, y)}}{m(b_{t-1}, o_t; y^{(1)}, \cdots y^{(K)})}\Big] = \mathbb{E}_{\mathbf{x}_1}\Big[\frac{\frac{1}{K} \sum_{j=1}^{K} e^{\mathcal{E}_\theta(b_{t-1}, o_t^{(1)}, y^{(j)})}}{m(b_{t-1}, o_t^{(1)}; y^{(1)}, \cdots y^{(K)})}\Big] = 1. \tag{14}$$

By applying Eq. (14) back to Eq. (13) and averaging the bound over $K$ samples, (reindexing $x^{(1)}$ as $x^{(j)}$ for each term), we exactly recover the lower bound on mutual information proposed by Oord et al. (2018) as:

$$1 + \mathbb{E}_{\mathbf{x}^{(j)}, y^{(j)}}\Big[\log \frac{e^{\mathcal{E}_\theta(b_{t-1}^{(j)}, y^{(j)})}}{a(b_{t-1}^{(j)}; y^{(1)}, \cdots, y^{(K)})}\Big] - \mathbb{E}_{\mathbf{x}^{(j)}}\Big[\frac{\mathbb{E}_{y^{(j)}} e^{\mathcal{E}_\theta(b_{t-1}^{(j)}, o_t^{(j)}, \hat{y}^{(j)})}}{a(b_{t-1}^{(j)}, o_t^{(j)}; y^{(1)}, \cdots, {}^{(K)})}\Big]$$

$$= \mathbb{E}_{\mathbf{x}^{(j)}, y^{(j)}}\Big[\log \frac{e^{\mathcal{E}_\theta(b_{t-1}^{(j)}, o_t^{(j)}, y^{(j)})}}{a(b_{t-1}^{(j)}, o_t^{(j)}; y^{(1)}, \cdots, y^{(K)})}\Big]$$

$$= \mathbb{E}_{\mathbf{x}^{(j)}, y^{(j)}}\Big[\log \frac{e^{\mathcal{E}_\theta(b_{t-1}^{(j)}, o_t^{(j)}, y^{(j)})}}{\frac{1}{K} \sum_{\hat{y}^{(j)}} e^{\mathcal{E}_\theta(b_{t-1}^{(j)}, o_t^{(j)}, \hat{y}^{(j)})}}\Big].$$

By multiplying and dividing the integrand in Eq. (9) by the $\frac{p(y|b_{t-1}^{(j)}, o_{t-1})}{Z(b_{t-1}^{(j)}, o_t)}$ and extracting $\frac{1}{K}$ out of the brackets, it transforms into:

$$\mathbb{E}_{\mathbf{x}^{(j)}, y^{(j)}}\Big[\log \frac{q_\theta(b_{t-1}^{(j)}, o_t^{(j)}, y^{(j)})}{\sum_{\hat{y}^{(j)}} q_\theta(b_{t-1}^{(j)}, o_t^{(j)}, \hat{y}^{(j)})}\Big] + \log(K).$$

$\square$

## A.2 Mutual Information and Conditional Entropy

Given scene annotation $y$, we measure the uncertainty of annotation $y$ at time step $t$ with conditional entropy of $y$ given series of observations $\{b_{t-1}, o_t\}$, denoted as $H(y|b_{t-1}, o_t)$. In this section, we prove that the conditional entropy decrease is equivalent to the conditional mutual information in Eq. (6).

**Theorem A.1.** *For any time step $t > 1$, the following holds:*

$$H(y|b_{t-1}) - H(y|b_{t-1}, o_t) = I(o_t; y|b_{t-1}). \tag{15}$$

*Proof.* Initially, by replacing the conditional entropy with the difference between entropy and mutual information, the left-hand side becomes:

$$H(y|b_{t-1}) - H(y|b_{t-1}, o_t)$$
$$= [H(y) - I(y; b_{t-1})] - [H(y) - I(y; b_{t-1}, o_t)]$$
$$= I(y; b_{t-1}, o_t) - I(y; b_{t-1}).$$

By applying Kolmogorov identities (Polyanskiy & Wu, 2014), it transforms into:

$$I(y; b_{t-1}, o_t) - I(y; b_{t-1}) = I(o_t; y|b_{t-1}).$$

With Theorem A.1, we can establish a connection between the mutual information in Eq. (6). The greedy informative exploration in Definition 3.1, thereby deducing the relationship between the policy model of EAD with the InfoNCE objective in Eq. (5) and greedy informative exploration.

## B OVERVIEW OF REINFORCEMENT LEARNING FUNDAMENTALS

At its core, reinforcement learning entails an agent learning to make decisions, interacting with its environment, and employing a Markov Decision Process (MDP) to model this interactive process. In the context of embodied perception, only three elements are relevant: observation, action, and state. The agent is in a particular state and needs to obtain observations from the environment to better understand it, take actions, interact with the environment, and transition to the next environment. To illustrate this with a robot, consider the procedure in which the robot captures an image from one specific position and then moves to the next position.

Furthermore, the Embodied Active Defense (EAD) method extends these concepts in reinforcement learning. Unlike making a one-time decision based on a single observation, EAD consistently monitors its environment, adjusting its understanding over time. It can be likened to a security camera that doesn't capture a single snapshot but rather continuously observes and adapts to its surroundings. A distinctive feature of EAD is its integration of perception (seeing and understanding) with action (taking informed actions based on that understanding). EAD doesn't merely passively observe; it actively interacts with its environment. This proactive engagement enhances EAD's ability to acquire more precise and reliable information, ultimately resulting in more informed decision-making.

## C EXPERIMENT DETAILS FOR SIMULATION ENVIRONMENT

**Envrionmental dynamics.** Formally, we define the state $s_t = (h_t, v_t)$, as a combination of camera's yaw $h_t \in \mathbb{R}$ and pitch $y_t \in \mathbb{R}$ at moment $t$, while the action is defined as continuous rotation denoted by $a_t = (\Delta h, \Delta v)$. Thereby, the transition function is denoted as $T(s_t, a_t, \mathbf{x}) = s_t + a_t$, while the observation function is reformulated with the 3D generative model $O(s_t, \mathbf{x}) = \mathcal{R}(s_t, \mathbf{x})$, where $\mathcal{R}(\cdot)$ is a renderer (*e.g.*, 3D generative model or graphic engine) that renders 2D image observation $o_t$ given camera parameters determined by state $s_t$.

In practice, the detailed formulation for renderer in computational graphics is presented as:

$$o_t = \mathcal{R}'(\boldsymbol{E}_t, \boldsymbol{I}, \mathbf{x}), \tag{16}$$

where $\boldsymbol{E}_t \in \mathbb{R}^{4\times4}$ is the camera's extrinsic determined by state $s_t$, while $\boldsymbol{I} \in \mathbb{R}^{3\times3}$ is the pre-defined camera intrinsic. That is to say, we need to calculate the camera's extrinsic $\boldsymbol{E}_t \in \mathbb{R}^{4\times4}$ with $s_t$ to utilize renderer to render 2D images. Assuming that we're using a right-handed coordinate system and column vectors, we have:

$$\boldsymbol{E}_t = \begin{bmatrix} \boldsymbol{R}_t & \boldsymbol{T} \\ \boldsymbol{0} & 1 \end{bmatrix}, \tag{17}$$

Where $\boldsymbol{R}_t \in \mathbb{R}^{3\times3}$ is the rotation matrix determined by $s_t$ and $\boldsymbol{T} \in \mathbb{R}^{3\times1}$ is the invariant translation vector. The rotation matrices for yaw $h_t$ and pitch $v_t$ are:

$$\boldsymbol{R}^y(h_t) = \begin{bmatrix} \cos(h_t) & 0 & \sin(h_t) \\ 0 & 1 & 0 \\ -\sin(h_t) & 0 & \cos(h_t) \end{bmatrix}, \tag{18}$$

$$\boldsymbol{R}^x(v_t) = \begin{bmatrix} 1 & 0 & 0 \\ 0 & \cos(v_t) & -\sin(v_t) \\ 0 & \sin(v_t) & \cos(v_t) \end{bmatrix}. \tag{19}$$

The combined rotation $R_t$ would then be $R_t = R^y(h_t) \times R^x(v_t)$. Then, the complete extrinsic matrix becomes:

$$E_t = \begin{bmatrix} \boldsymbol{R}^y(h_t) \times \boldsymbol{R}^x(v_t) & \boldsymbol{T} \\ \boldsymbol{0} & 1 \end{bmatrix}. \tag{20}$$

**Applying function.** In our experiments, the adversarial patch is attached to a flat surface, such as eyeglasses for face recognition and billboards for object detection. Utilizing known corner coordinates of the adversarial patch in the world coordinate system, we employ both extrinsic $\boldsymbol{E}_t$ and intrinsic $\boldsymbol{K}$ parameters to render image observations containing the adversarial patch. We follow the projection process of 3D patch by Zhu et al. (2023) to construct the applying function, with the projection matrix $\mathcal{M}_{3d-2d} \in \mathbb{R}^{4 \times 4}$ specified as follows:

$$\mathcal{M}_{3d-2d} = \begin{bmatrix} \boldsymbol{K} & \boldsymbol{0} \\ \boldsymbol{0} & 1 \end{bmatrix} \times \boldsymbol{E}_t. \tag{21}$$

This process is differentiable, which allows for the optimization of the adversarial patches.

In summary, we propose a deterministic environmental model applicable to all our experimental environments (*e.g.*, EG3D, CARLA):

$$
\begin{aligned}
\text{State} \quad & s_t = (h_t, v_t) \in \mathbb{R}^2, \\
\text{Action} \quad & a_t = (\Delta h, \Delta v) \in \mathbb{R}^2, \\
\text{Transition Function} \quad & T(s_t, a_t, \mathbf{x}) = s_t + a_t, \\
\text{Observation Function} \quad & Z(s_t, \mathbf{x}) = \mathcal{R}(s_t, \mathbf{x}).
\end{aligned}
$$

The primary distinction between simulations for different tasks lies in the feasible viewpoint regions. These are detailed in the implementation sections for each task, specifically in Appendices D and E.

## D  EXPERIMENT DETAILS FOR FACE RECOGNITION

### D.1  CELEBA-3D

We use unofficial implemented GAN Inversion with EG3D (https://github.com/oneThousand1000/EG3D-projector) and its default parameters to convert 2D images from CelebA (Liu et al., 2018) into 3D latent representation $w^+$. For the 3D generative model prior, we use the EG3D models pre-trained on FFHQ which is officially released at https://catalog.ngc.nvidia.com/orgs/nvidia/teams/research/models/eg3d. For lower computational overhead, we desert the super-resolution module of EG3D and directly render RGB images of $112 \times 112$ with its neural renderer.

We further evaluate the quality of the reconstructed CelebA-3D dataset. We measure image quality with PSNR, SSIM and LPIPS between original images and EG3D-rendered images from the same viewpoint. We later evaluate the identity consistency between the reconstructed 3D face and their original 2D face with identity consistency (ID), which is slightly different from the one in (Chan et al., 2022), for we measure them by calculating the mean ArcFace (Deng et al., 2019) cosine similarity score between pairs of views of the face rendered from random camera poses and its original image from CelebA.

As shown in Table 4, the learned 3D prior over FFHQ enables surprisingly high-quality single-view geometry recovery. So our reconstructed CelebA-3D is equipped with high image quality and sufficient identity consistency with its original 2D form for later experiments. And the selection of reconstructed multi-view faces is shown in Figure 5.

The CelebA-3D dataset inherits annotations from the original CelebA dataset, which is accessible at https://mmlab.ie.cuhk.edu.hk/projects/CelebA.html. The release of this dataset for public access is forthcoming.

Table 4: quantitative evaluation for CelebA-3D. The image size is $112 \times 112$.

|  | PSNR↑ | SSIM↑ | LPIPS↓ | ID↑ |
| --- | --- | --- | --- | --- |
| CelebA-3D | 21.28 | .7601 | .1314 | .5771 |

Original                                   Reconstruction

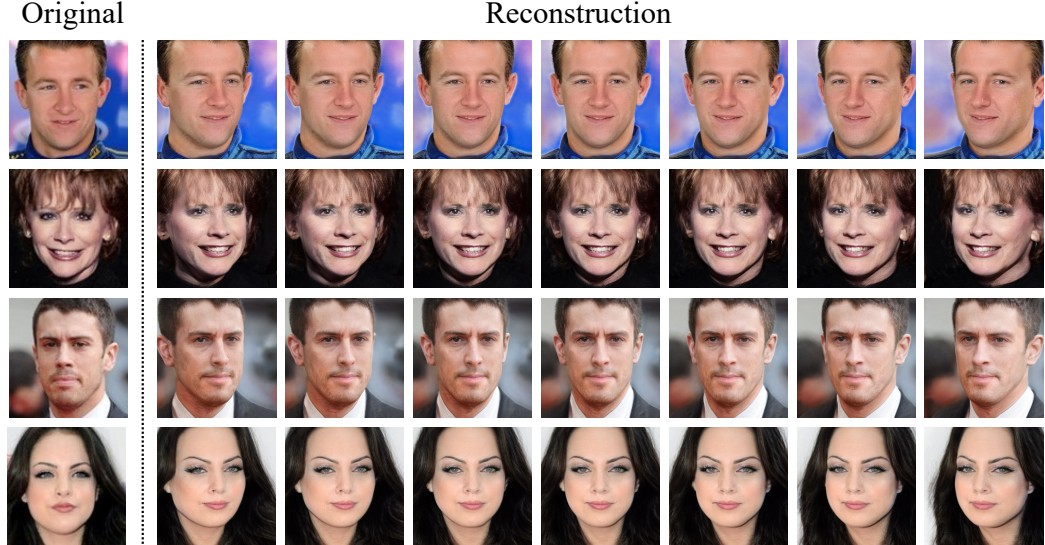

Figure 5: qualitative evaluation for CelebA-3D. The first column presents the original face from CelebA, and the subsequent columns demonstrate the rendered multiview faces from inverted $w^+$ with EG3D. The image size is $112 \times 112$.

### D.2 DETAILS FOR ATTACKS

**Impersonation and dodging in adversarial attacks against FR system.** In adversarial attacks against FR systems, two main subtasks are identified: impersonation and dodging. Impersonation involves manipulating an image to deceive FR systems into misidentifying an individual as another specific person. Dodging, conversely, aims to prevent accurate identification by the FR system. It involves modifying an image so that the FR system either fails to detect a face or cannot associate the detected face with any data of its authentic identity in its database. These subtasks pose significant challenges to FR system security, highlighting the need for robust countermeasures in FR technology.

**Attacks in pixel space.** The Momentum Iterative Method (MIM) (Dong et al., 2018) and Expectation over Transformation (EoT) (Athalye et al., 2018b) focus on refining adversarial patches in RGB pixel space. MIM enhances the generation of adversarial examples by incorporating momentum into the optimization process, which enhances transferring of adversarial examples. In contrast, EoT considers various transformations, such as rotations and lighting changes, thereby strengthening the robustness of attacks under diverse physical conditions. For implementation, we adhered to the optimal parameters as reported in (Xiao et al., 2021), setting the number of iterations at $N = 150$, the learning rate at $\alpha = 1.5/255$, and the decay factor at $\mu = 1$. These parameters were consistent across all experiments. The sampling frequency for EoT was established at $M = 10$.

**Attacks in latent space of the generative model.** In the context of GenAP (Xiao et al., 2021) and Face3DAdv (Yang et al., 2022), the focus was on optimizing adversarial patches within the latent space of a Generative Adversarial Network (GAN). GenAP utilizes the generative capabilities of GANs to develop adversarial patches, whereas Face3DAdv is specifically designed for attacking facial recognition systems, accounting for 3D variations. Additionally, we employed the latent space of EG3D for patch optimization, opting for the Adam optimizer (Kingma & Ba, 2014) with a learning rate of $\eta = 0.01$ and an iteration count of $N = 150$. The sampling frequency was also set at $M = 10$.

### D.3 DETAILS FOR ADAPTIVE ATTACKS

**Adaptive attack for defense baselines.** To launch adaptive attacks against parameter-free, purification-based defenses such as JPEG and LGS, we employ Backward Pass Differentiable Approximation (BPDA) as proposed by Athalye et al. (2018a). This method assumes that the output from each defense mechanism closely approximates the original input. For adaptive attacks on SAC and PZ, we use their official implementations (Liu et al., 2022), incorporating Straight-Through Estimators (STE) (Bengio et al., 2013) for backpropagation through thresholding operations.

**Adaptive attack with uniform superset policy.** In adaptive attacks for EAD, we leverage uniform superset approximation for the policy model. Thus, we have the surrogate policy

$$\tilde{\pi} := \mathcal{U}(h_{\min}, h_{\max}) \times \mathcal{U}(v_{\min}, v_{\max}), \tag{22}$$

where $a_t \in [h_{\min}, h_{\max}] \times [v_{\min}, v_{\max}]$, and $[h_{\min}, h_{\max}]$, $[v_{\min}, v_{\max}]$ separately denotes the pre-defined feasible region for horizontal rotation (yaw) and vertical rotation (pitch). The optimization objective is outlined as follows, with a simplified sequential representation for clarity[2]:

$$
\begin{aligned}
\max_p \quad & \mathbb{E}_{s_1 \sim P_1, a_i \sim \tilde{\pi}} \mathcal{L}(\overline{y}_\tau, y), \\
\text{with} \quad & \{\overline{y}_\tau, b_\tau\} = f(\{A(o_i, p; s_1 + \sum_{j=1}^{i-1} a_j)\}_{i=1}^\tau; \boldsymbol{\theta}) \\
\text{s.t.} \quad & p \in [0, 1]^{H_p \times W_p \times C},
\end{aligned}
\tag{23}
$$

where $P_1$ denotes the distribution of initial state $s_1$, and $\mathcal{L}$ is the task-specific loss function.

**Adaptive Attack Against Sub-Modules.** An end-to-end attack may not always be the most effective strategy, particularly against defenses with complex forward passes. Targeting the weakest component is often sufficient. Therefore, we propose two separate adaptive attacks: one against the perception model and another against the policy model. The attack on the perception model aims to generate an adversarial patch that corrupts the internal belief $b_t$ (Sabour et al., 2015). The optimization objective for this attack is to maximize the Euclidean distance between the corrupted belief $b\tau$ and the benign belief $b\tau^+$, formulated as follows:

$$
\begin{aligned}
\max_p \quad & \mathbb{E}_{s_1 \sim P_1, a_i \sim \tilde{\pi}} \|b_\tau - b_\tau^+\|_2^2, \\
\text{with} \quad & \{\overline{y}_\tau, b_\tau\} = f(\{A(o_i, p; s_1 + \sum_{j=1}^{i-1} a_j)\}_{i=1}^\tau; \boldsymbol{\theta}), \\
& \{\overline{y}_\tau^+, b_\tau^+\} = f(\{o_i\}_{i=1}^\tau; \boldsymbol{\theta}), \\
\text{s.t.} \quad & p \in [0, 1]^{H_p \times W_p \times C}.
\end{aligned}
\tag{24}
$$

For the attack against the policy model, the goal is to create an adversarial patch that induces the policy model to output a zero action $a_i = \pi(b_i; \phi) = 0$, thereby keeping the EAD model stationary with an invariant state $s_i = s_1$ and generating erroneous predictions $\overline{y}_\tau$. While the original problem can be challenging with policy output as a constraint, we employ Lagrangian relaxation to incorporate the constraint into the objective and address the following problem:

$$
\begin{aligned}
\max_p \quad & \mathbb{E}_{s_1 \sim P_1} \mathcal{L}(\overline{y}_\tau, y) + c \cdot \|\pi(\{A(o_1, p; s_1)\}_{i=1}^\tau; \phi)\|_2^2, \\
\text{with} \quad & \{\overline{y}_\tau, b_\tau\} = f(\{A(o_1, p; s_1)\}_{i=1}^\tau; \boldsymbol{\theta}) \\
\text{s.t.} \quad & p \in [0, 1]^{H_p \times W_p \times C},
\end{aligned}
\tag{25}
$$

where $c > 0$ is a constant that yields an adversarial example ensuring the model outputs zero actions.

---

[2] The recurrent inference procedure is presented sequentially in this section for simplicity.

**Adaptive Attack for the Entire Pipeline.** Attacking the EAD model through backpropagation is infeasible due to the rapid consumption of GPU memory as trajectory length increases (*e.g.*, 4 steps require nearly 90 GB of video memory). To mitigate this, we use gradient checkpointing (Chen et al., 2016) to reduce memory consumption. By selectively recomputing parts of the computation graph defined by the $\tau$-step EAD inference procedure, instead of storing them, this technique effectively reduces memory costs at the expense of additional computation. Using this method, we successfully attacked the entire pipeline along a 4-step trajectory using an NVIDIA RTX 3090 Ti.

Regarding implementation, we adopted the same hyper-parameters as Face3DAdv and considered the action bounds defined in Appendix D.5. For the constant $c$ in the adaptive attack against the policy model, we employed a bisection search to identify the optimal value as per Carlini & Wagner (2017), finding $c = 100$ to be the most effective. Additionally, the evaluation results from these adaptive attacks and analysis are detailed in Appendix D.7 with main results in Table 9.

## D.4 DETAILS FOR DEFENSES

**JPEG compression.** we set the quality parameter to 75.

**Local gradients smoothing.** We adopt the implementation at `https://github.com/fabiobrau/local_gradients_smoothing`, and maintain the default hyper-parameters claimed in Naseer et al. (2019).

**Segment and complete.** We use official implementation at `https://github.com/joellliu/SegmentAndComplete`, and retrain the patch segmenter with adversarial patches optimized by EoT (Athalye et al., 2018a) and USAP technique separately. We adopt the same hyper-parameters and training process claimed in Liu et al. (2022), except for the prior patch sizes, which we resize them proportionally to the input image size.

**Patchzero.** For Patchzero (Xu et al., 2023), we directly utilize the trained patch segmenter of SAC, for they share almost the same training pipeline.

**Defense against Occlusion Attacks.** DOA is an adversarial training-based method. We adopt the DOA training paradigm to fine-tune the same pre-trained IResNet-50 and training data as EAD with the code at `https://github.com/P2333/Bag-of-Tricks-for-AT`. As for hyper-parameters, we adopt the default ones in Wu et al. (2019) with the training patch size scaled proportionally.

## D.5 DETAILS FOR IMPLEMENTATIONS

**Model details.** In the context of face recognition, we implement EAD model with a composition of a pre-trained face recognition feature extractor and a Decision transformer, specifically, we select IResNet-50 as the the visual backbone to extract feature. For each time step $t > 0$, we use the IResNet-50 to map the current observation input of dimensions $112 \times 112 \times 3$ into an embedding with a dimensionality of $512$. This embedding is then concatenated with the previously extracted embedding sequence of dimensions $(t-1) \times 512$, thus forming the temporal sequence of observation embeddings ($t \times 512$) for Decision Transformer input. The Decision Transformer subsequently outputs the temporal-fused face embedding for inference as well as the predicted action. For the training process, we further map the fused face embedding into logits using a linear projection layer. For the sake of simplicity, we directly employ the Softmax loss function for model training.

In experiments, we use pre-trained IResNet-50 with ArcFace margin on MS1MV3 (Guo et al., 2016) from `InsightFace` (Deng et al., 2019), which is available at `https://github.com/deepinsight/insightface/tree/master/model_zoo`.

**Phased training.** It's observed that the training suffers from considerable instability when simultaneously training perception and policy models from scratch. A primary concern is that the perception model, in its early training stages, is unable to provide accurate supervision signals, leading the policy network to generate irrational actions and hindering the overall learning process. To mitigate this issue, we initially train the perception model independently using frames obtained from a random

action policy, namely *offline phase*. Once achieving a stable performance from the perception model, we proceed to the *online phase* and jointly train both the perception and policy networks, employing Algorithm 1,thereby ensuring their effective coordination and learning.

Meanwhile, learning offline with pre-collected data in the first phase proves to be significantly more efficient than online learning through interactive data collection from the environment. By dividing the training process into two distinct phases *offline* and *online*, we substantially enhance training efficiency and reduce computational costs.

**Training details.** To train EAD for face recognition, we randomly sample images from $2,500$ distinct identities from the training set of CelebA-3D. we adopt the previously demonstrated phased training paradigm with hyper-parameters listed in Table 5.

Table 5: Hyper-parameters of EAD for face recognition

| Hyper-parameter | Value |
|---|---|
| Lower bound for horizontal rotation ($h_{min}$) | $-0.35$ |
| Upper bound for horizontal rotation ($h_{max}$) | $0.35$ |
| Lower bound for vertical rotation ($v_{min}$) | $-0.25$ |
| Upper bound for vertical rotation ($v_{max}$) | $0.25$ |
| Ratio of patched data ($r_{patch}$) | $0.4$ |
| Training epochs for offline phase ($lr_{offline}$) | $50$ |
| learning rate for offline phase ($lr_{offline}$) | 1E-3 |
| batch size for offline phase ($b_{offline}$) | $64$ |
| Training epochs for online phase | $50$ |
| learning rate for online phase ($lr_{online}$) | 1.5E-4 |
| batch size for offline phase ($b_{online}$) | $48$ |

## D.6 COMPUTATIONAL OVERHEAD

This section evaluates our method's computational overhead compared to other passive defense baselines in facial recognition systems. The performance assessment is conducted on a NVIDIA GeForce RTX 3090 Ti and an AMD EPYC 7302 16-Core Processor, using a training batch size of 64. SAC and PZ necessitate training a segmenter to identify the patch area, entailing two stages: initial training with pre-generated adversarial images and subsequent self-adversarial training (Liu et al., 2022; Xu et al., 2023). DOA, an adversarial training-based approach, requires retraining the feature extractor (Wu et al., 2019). Additionally, EAD's training involves offline and online phases, without involving adversarial training.

As indicated in Table 6, although differential rendering imposes significant computational demands during the online training phase, the total training time of our EAD model is effectively balanced between the pure adversarial training method DOA and the partially adversarial methods like SAC and PZ. This efficiency stems mainly from our unique USAP approach, which bypasses the need for generating adversarial examples, thereby boosting training efficiency. In terms of model inference, our EAD, along with PZ and DOA, demonstrates superior speed compared to LGS and SAC. This is attributed to the latter methods requiring CPU-intensive, rule-based image preprocessing, which diminishes their inference efficiency.

Regarding detailed training, the EAD model was trained following the configuration in Appendix D.5. The offline training utilized 2 NVIDIA Tesla A100 GPUs for approximately 4 hours (210 minutes). Due to the substantial memory demands of EG3D differential rendering, the online training phase required 8 NVIDIA Tesla A100 GPUs and extended to about 14 hours (867 minutes). Figure 6 illustrates the training curves of our method.

Table 6: omputational overhead comparison of different defense methods in face recognition. We report the training and inference time of defense on a NVIDIA GeForce RTX 3090 Ti and an AMD EPYC 7302 16-Core Processor with the training batch size as 64.

| Method | # Params (M) | Parametric Model | Training Epochs | Training Time per batch (s) | Overall Training Time (GPU hours) | Inference Time per Instance (ms) |
|---|---|---|---|---|---|---|
| JPEG | - | non-parametric | - | - | - | 9.65 |
| LGS | - | non-parametric | - | - | - | 26.22 |
| SAC | 44.71 | segmenter | $50 + 10$ | $0.152/4.018$ | 104 | 26.43 |
| PZ | | | | | | 11.88 |
| DOA | 43.63 | feature extractor | 100 | 1.732 | 376 | 8.10 |
| EAD | 57.30 | policy and perception model | $50 + 50$ | $0.595/1.021$ | 175 | 11.51 |

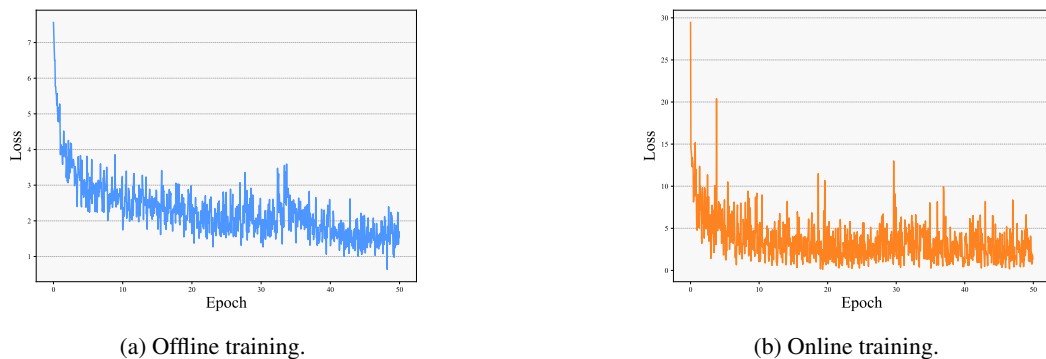

(a) Offline training.

(b) Online training.

Figure 6: The training curves of EAD model for face recognition.

## D.7   MORE EVALUATION RESULTS

**Evaluation with different patch sizes.**   To further assess the generalizability of the EAD model across varying patch sizes and attack methods, we conduct experiments featuring both impersonation and doding attacks. These attacks share similarities with the setup illustrated in Table 1. Although with different patch sizes, the results in Table 7 and Table 8 bear a considerable resemblance to those displayed in Table 1. This congruence further supports the adaptability of the EAD model in tackling unseen attack methods and accommodating diverse patch sizes.

**Evaluation with different attack iterations.**   Figure 7 shows that EAD generalizes well under a wide spectrum of attack iterations, and the only ASR slightly increases when the patch gets larger. As for dodging attacks, Figure 8 demonstrates EAD's superior generalizability compared to other defense strategies when countering dodging attacks. This is akin to the observations in Figure 3, where EAD, trained exclusively with patches comprising 10% of the image, consistently exhibits a low attack success rate, even with increasing patch sizes and attack iterations. This further underscores EAD's exceptional generalizability. One subtle phenomenon to note is that the segmenter-based method (*i.e.* SAC and PZ) degrades when the attack iterations become smaller. Because the patches become more imperceptible.

**Evaluation with different adaptive attacks.**   As Table 9 demonstrates, our original adaptive attack using USP was more effective than tracing the authentic policy of EAD (overall). This may be

Table 7: The white-box impersonation attack success rates on face recognition models with different patch sizes. [†] denotes methods are trained with adversarial examples.

| Method | 8% | | | | 10 % | | | | 12% | | | |
|---|---|---|---|---|---|---|---|---|---|---|---|---|
| | MIM | EoT | GenAP | 3DAdv | MIM | EoT | GenAP | 3DAdv | MIM | EoT | GenAP | 3DAdv |
| Undefended | 100.0 | 100.0 | 99.00 | 98.00 | 100.0 | 100.0 | 100.0 | 99.00 | 100.0 | 100.0 | 100.0 | 99.00 |
| JPEG | 99.00 | 100.0 | 99.00 | 93.00 | 100.0 | 100.0 | 99.00 | 99.00 | 100.0 | 100.0 | 99.00 | 99.00 |
| LGS | 5.10 | 7.21 | 33.67 | 30.61 | 6.19 | 7.29 | 41.23 | 36.08 | 7.21 | 12.37 | 61.85 | 49.48 |
| SAC | 6.06 | 9.09 | 67.68 | 64.64 | **1.01** | 3.03 | 67.34 | 63.26 | 5.05 | 4.08 | 69.70 | 66.32 |
| PZ | 4.17 | 5.21 | 59.38 | 45.83 | 2.08 | 3.13 | 60.63 | 58.51 | 4.17 | **3.13** | 60.63 | 58.33 |
| SAC[†] | 3.16 | 3.16 | 18.94 | 22.11 | 2.10 | 3.16 | 21.05 | 16.84 | 3.16 | 4.21 | 15.78 | 18.95 |
| PZ[†] | **3.13** | 3.16 | 19.14 | 27.37 | 2.11 | 3.13 | 20.00 | 30.53 | 5.26 | 5.26 | 18.95 | 28.42 |
| DOA[†] | 95.50 | 89.89 | 96.63 | 89.89 | 95.50 | 93.26 | 100.0 | 96.63 | 94.38 | 93.26 | 100.0 | 100.0 |
| **EAD (ours)** | 4.12 | **3.09** | **5.15** | **7.21** | 3.09 | **2.06** | **4.17** | **8.33** | **3.09** | 5.15 | **8.33** | **10.42** |

Table 8: The white-box dodging attack success rates (%) on face recognition models with different patch sizes. [†] denotes methods are trained with adversarial examples.

| Method | 8% | | | | 10 % | | | | 12% | | | |
|---|---|---|---|---|---|---|---|---|---|---|---|---|
| | MIM | EoT | GenAP | 3DAdv | MIM | EoT | GenAP | 3DAdv | MIM | EoT | GenAP | 3DAdv |
| Undefended | 100.0 | 100.0 | 99.00 | 89.00 | 100.0 | 100.0 | 100.0 | 95.00 | 100.0 | 100.0 | 100.0 | 99.00 |
| JPEG | 98.00 | 99.00 | 95.00 | 88.00 | 100.0 | 100.0 | 99.00 | 95.00 | 100.0 | 100.0 | 100.0 | 98.00 |
| LGS | 49.47 | 52.63 | 74.00 | 77.89 | 48.93 | 52.63 | 89.47 | 75.78 | 55.78 | 54.73 | 100.0 | 89.47 |
| SAC | 73.46 | 73.20 | 92.85 | 78.57 | 80.06 | 78.57 | 92.85 | 91.83 | 76.53 | 77.55 | 92.85 | 92.92 |
| PZ | 6.89 | 8.04 | 58.44 | 57.14 | 8.04 | 8.04 | 60.52 | 65.78 | 13.79 | 12.64 | 68.49 | 75.71 |
| SAC[†] | 78.78 | 78.57 | 79.59 | 85.85 | 81.65 | 80.80 | 82.82 | 86.73 | 80.61 | 84.69 | 87.87 | 87.75 |
| PZ[†] | 6.12 | 6.25 | 14.29 | 20.41 | 7.14 | 6.12 | 21.43 | 25.51 | 11.22 | 10.20 | 24.49 | **30.61** |
| DOA[†] | 75.28 | 67.42 | 87.64 | 95.51 | 78.65 | 75.28 | 97.75 | 98.88 | 80.90 | 82.02 | 94.38 | 100.0 |
| **EAD (ours)** | **0.00** | **0.00** | **2.10** | **13.68** | **2.11** | **1.05** | **6.32** | **16.84** | **2.10** | **3.16** | **12.64** | 34.84 |

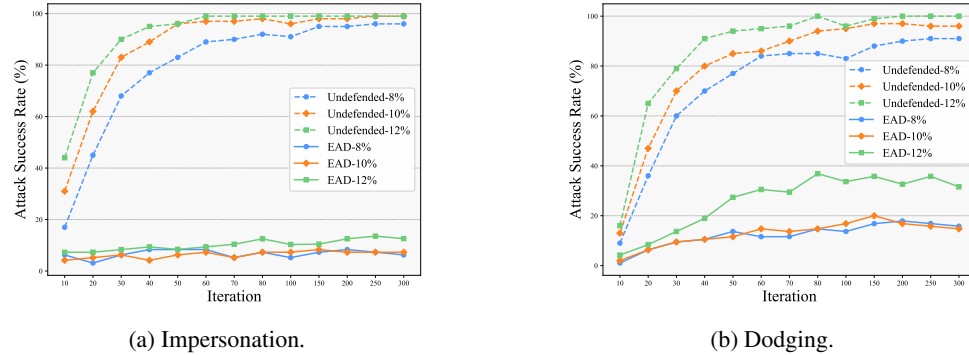

(a) Impersonation.  (b) Dodging.

Figure 7: Performance of EAD under varying attack iterations and patch sizes. The adversarial patches are crafted by 3DAdv.

attributed to vanishing or exploding gradients (Athalye et al., 2018a) that impedes optimization. This problem is potentially mitigated by our approach of computing expectations over a uniform policy distribution. In the meantime, The results reaffirm the robustness of EAD against a spectrum of adaptive attacks. It further shows that EAD's defensive capabilities arise from the synergistic integration of its policy and perception models, facilitating strategic observation collection rather than learning a short-cut strategy to neutralize adversarial patches from specific viewpoints.

**Impact of horizon length.** We later demonstrate how the horizon length (*i.e.*, decision steps) influence the performance of EAD in Figure 9. Two key observations emerge from this analysis: 1) The standard accuracy initially exhibits an increase but subsequently declines as $t$ surpasses the maximum trajectory length $\tau = 4$ which is manually set in model training, as detailed in Figure 9a. 2)

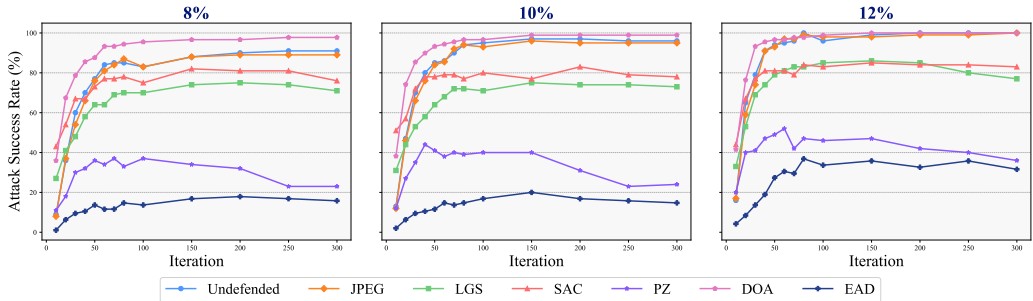

Figure 8: Comparative evaluation of defense methods across varying attack iterations with different adversarial patch sizes. The sizes of the adversarial patches for each subfigure, from left to right, are 8%, 10%, and 12%, respectively. The adversarial patches are crafted by 3DAdv for dodging.

Table 9: Evaluation of adaptive attacks on the EAD Model. Columns with *USP* represent results obtained by optimizing the patch with expected gradients over the Uniform Superset Policy (USP). And *perception* and *policy* separately represent adaptive attack against single sub-module. And *overall* denotes attacking EAD by following the gradients for along the overall (4 step) trajectory with gradient-checkpointing.

| | Dodging | | | | Impersonation | | | |
|---|---|---|---|---|---|---|---|---|
| | **USP** | **Perception** | **Policy** | **Overall** | **USP** | **Perception** | **Policy** | **Overall** |
| **ASR (%)** | **22.11** | 10.11 | 16.84 | 15.79 | 8.33 | 1.04 | **9.38** | 7.29 |

The changes in the similarity between face pairs affected by adversarial patches consistently decline. Specifically, for impersonation attacks (Figure 9b), the change in similarity implies an increase, while a decrease is noted for dodging attacks (Figure 9c). This trend indicates that more information from additional viewpoints is accumulated during the decision process, effectively mitigating the issues of information loss and model hallucination caused by adversarial patches.

**Efficiency of EAD's policy.** Although the efficiency of EAD's policy is theoretically demonstrated as it has proven to be a greedy informative strategy in Sec. 3.2, we further validate the superiority of the EAD's policy empirically by comparing the performance of EAD and EAD integrated with a random movement policy, hereafter referred to as $EAD_{RAND}$. Both approaches share identical neural network architecture and parameters. Figures 9b and 9c illustrate that the $EAD_{RAND}$ is unable to mitigate the adversarial effect with random exploration, even when more action are employed. Consequently, the exploration efficiency of the random policy significantly lags behind the approximate solution of the greedy informative strategy, which is derived from parameter learning.

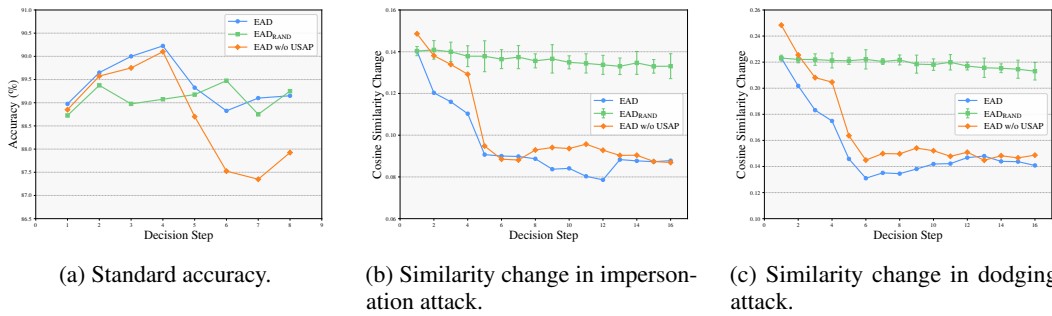

(a) Standard accuracy.    (b) Similarity change in impersonation attack.    (c) Similarity change in dodging attack.

Figure 9: Model's performance variation along the number of decision steps.

**Evaluation under various viewpoints.** To further assess the versatility of defenses over multiple viewpoints, we examine model performance under 3D viewpoint variation. For all the experiments, we utilize facial images requiring specific movement ranges for the cruciform rail, spanning from

$-15°$ to $15°$, corresponding to *yaw* and *pitch* respectively. Moreover, these conditions are linearly combined to form a new category termed as *mixture*. We employ Face3DAdv as the adversary, provided that solely adversarial examples crafted with Face3DAdv hat do not exhibit performance degradation when encountering viewpoint variations. Table 10 demonstrates that EAD consistently upholds superior performance, even under varied viewpoint conditions.

Table 10: Standard accuracy (%) and attack success rates (%) of Face3DAdv on face recognition models under various testing viewpoint protocols. "Acc." denotes the accuracy with best threshold calculated with standard protocol from LFW (Huang et al., 2007). "Imp." and "Dod." respectively represents the attack success rate of impersonation and dodging attacks.

| Method | Yaw | | | Pitch | | | Mixture | | |
|---|---|---|---|---|---|---|---|---|---|
| | Acc. | Imp. | Dod. | Acc. | Imp. | Dod. | Acc. | Imp. | Dod. |
| Undefended | 98.64 | 94.74 | 87.24 | 98.87 | 91.95 | 86.04 | 98.39 | 88.35 | 83.65 |
| JPEG | 98.88 | 91.48 | 86.43 | 98.94 | 89.26 | 84.83 | 98.39 | 84.94 | 82.60 |
| LGS | 93.12 | 29.32 | 72.39 | 92.42 | 26.23 | 69.32 | 91.92 | 24.54 | 68.50 |
| SAC | 95.47 | 57.22 | 75.21 | 96.30 | 59.83 | 74.84 | 95.24 | 53.83 | 74.04 |
| PZ | 97.81 | 41.07 | 51.91 | 97.77 | 39.99 | 49.66 | 97.39 | 36.35 | 48.54 |
| **EAD (ours)** | - | - | - | - | - | - | **99.28** | **13.68** | **7.21** |

## D.8 MORE QUALITATIVE RESULTS

**Qualitative comparison of different version of SAC.** SAC is a preprocessing-based method that adopts a segmentation model to detect patch areas, followed by a "shape completion" technique to extend the predicted area into a larger square, and remove the suspicious area (Liu et al., 2022). As shown in Figure 10, the enhanced SAC, while exhibiting superior segmentation performance in scenarios like face recognition, inadvertently increases the likelihood of masking critical facial features such as eyes and noses. This leads to a reduced ability of the face recognition model to correctly identify individuals, thus impacting its performance in dodging attacks.

Regular SAC 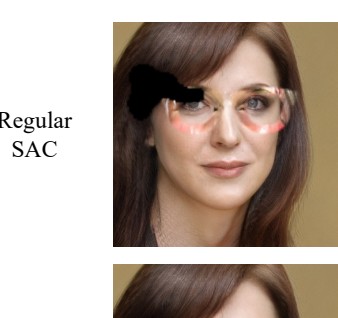 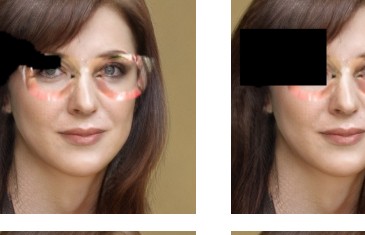 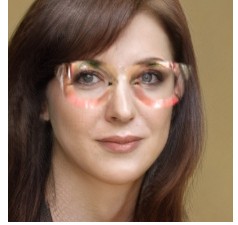

Enhanced SAC 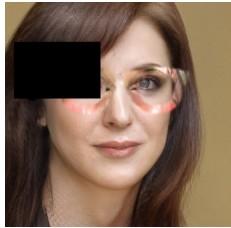 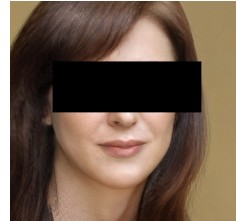

Adversarial Image

Image Processed with Segmenter-predicted Mask    Image Processed with Completed Mask

Figure 10: Qualitative results of SAC trained with different data. The first column present the adversarial image processed by regular SAC which trained with patch filled with Gaussian noise, while the subsequent column demonstrate the one processed by enhanced SAC. The adversarial patches are generated with 3DAdv and occupy 8% of the image.

**More qualitative results of EAD.** As shown in Figure 11, EAD model prioritizes a distinct viewpoint to improve target understanding while simultaneously maintaining a perspective where adversarial patches are minimally effective.

Step 1        Step 2        Step 3        Step 4

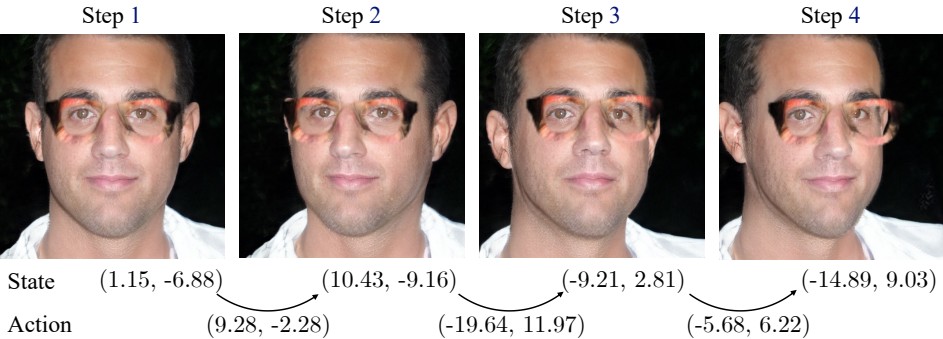

State    (1.15, -6.88)      (10.43, -9.16)      (-9.21, 2.81)      (-14.89, 9.03)

Action        (9.28, -2.28)        (-19.64, 11.97)        (-5.68, 6.22)

Figure 11: Qualitative results of EAD. The state represents the yaw and pitch of camera, and the action indicates the camera rotation predicted by EAD's policy model. The adversarial patches are generated with 3DAdv and occupy $8\%$ of the image.

# E EXPERIMENT DETAILS FOR OBJECT DETECTION

## E.1 DETAILS FOR EXPERIMENTAL DATA

**Details on CARLA.** For model testing, we use the scene basis `billboard05` from CARLA-GEAR (Nesti et al., 2022). For each testing scene, we randomly place different objects within the scene to ensure the diversity of testing scenarios. In this setting, the adversarial patch are affixed to billboards on the street. scene examples are represented in Figure 12.

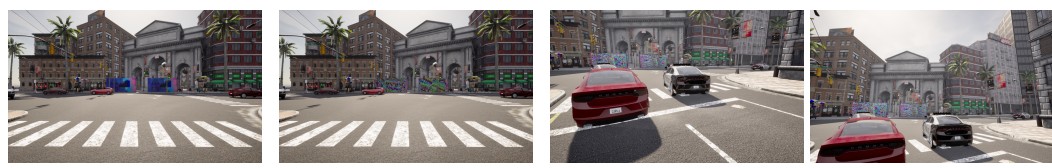

Figure 12: Scene examples from `billboard05` of CARLA-GEAR. The first two and last two images belong to different scenes. The first image is a benign image, while the second image is contaminated with adversarial patches attached to the billboards. For the last two images, they are rendered from different viewpoints.

**Details on EG3D.** We use a pre-trained EG3D model on ShapeNet Cars at https://catalog.ngc.nvidia.com/orgs/nvidia/teams/research/models/eg3d to generate multi-view car images with the resolution of $128 \times 128$. And the seed is $114514$. As for data annotation, We select the pre-trained YOLOv5x model which is the largest and performs the best (Jocher et al., 2021). We fine-tune it on the generated car images for $500$ iterations and then utilize it as an annotation model to label bounding boxes for later generated images.

## E.2 DETAILS FOR ATTACKS

As for MIM (Dong et al., 2018), we set the decay factor $\mu = 1.0$, while the size of the Gaussian kernel is 21 for TIM (Dong et al., 2019).

**Details on CARLA.** For attacks in CARLA, we utilize the attack against Mask-RCNN implemented by CARLA-GEAR at https://github.com/retis-ai/PatchAttackTool and its default hyper-parameters.

**Details on EG3D.** For attacks in environment powered by EG3D, we set the number of iterations as $N = 100$ and the learning rate $\alpha = 10/255$.

### E.3 DETAILS FOR DEFENSE

For SAC[†] and PZ[†], We use official implementation and its pre-trained patch segmenter checkpoint for object detection at https://github.com/joellliu/SegmentAndComplete, which is trained with adversarial examples generated with PGD (Madry et al., 2017).

### E.4 DETAILS FOR IMPLEMENTATION

**Model details.** For experiment conducted on simulation environment based on EG3D, We implement EAD for object detection with a combination of YOLOv5n and Decision Transformer. For each time step $t > 0$, given the current observation of dimensions $640 \times 640 \times 3$ as input, the smallest feature maps ($4 \times 4 \times 512$) within the feature pyramid are utilized. These maps are extracted via the YOLOv5 backbone and reshaped into a sequence with dimensions $16 \times 512$. we utilize the smallest feature maps ($4 \times 4 \times 512$) in the feature pyramid which is extracted with the YOLOv5 backbone and reshape it into a sequence ($16 \times 512$). To concatenate it with the previous extracted observation sequence $16(t - 1) \times 512$, we have a temporal sequence of visual features as the input of Decision Transformer, and it output the temporal-fused visual feature sequences ($16 \times 512$) and predicted action. To predict the bounding boxes and target label which is required in object detection, we reshape the temporal-fused visual sequence back to its original shape and utilize it as part of the feature pyramid in the later stage for object detection in YOLOv5n. As for experiment on CARLA, we utilize a combination of Mask-RCNN and Decision Transformer with similar implementation technique.

We use the official implementation and pre-trained model checkpoints for both YOLOv5n and YOLOv5x at https://github.com/ultralytics/yolov5. As for Mask-RCNN, we adopt the implementation from torchvision.

**Training details.** We adopt a similar training paradigm as EAD for face recognition and set the hyper-parameters of EAD for object detection as follows:

Table 11: Hyper-Parameters of EAD for object detection

| Hyper-Parameter | Value |
| --- | --- |
| Lower bound for horizontal rotation ($h_{\min}$) | $-\pi/2$ |
| Upper bound for horizontal rotation ($h_{\max}$) | $\pi/2$ |
| Lower bound for vertical rotation ($v_{\min}$) | 0 |
| Upper bound for vertical rotation ($v_{\max}$) | 0.2 |
| Ratio of patched data ($r_{\text{patch}}$) | 0.4 |
| Training iterations for offline phase | 500 |
| learning rate for offline phase ($\text{lr}_{\text{offline}}$) | 2E-5 |
| batch size for offline phase ($b_{\text{offline}}$) | 64 |
| Training iterations for online phase | 80 |
| learning rate for online phase ($\text{lr}_{\text{online}}$) | 1E-5 |
| batch size for online phase ($b_{\text{online}}$) | 32 |

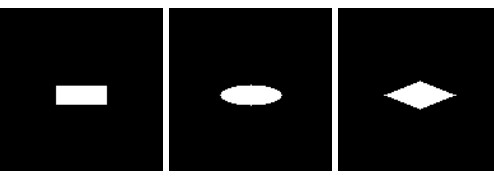

Figure 13: Different shapes used for evaluation. From left to right: rectangle, ellipse and diamond.

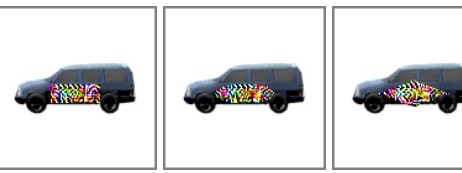

Figure 14: Adversarial examples with patches of different shapes. From left to right: rectangle, ellipse and diamond.

### E.5 MORE EVALUATION RESULTS ON EG3D

**Evaluation with different patch shapes.** Credits to other shapes of the patch are either too skeptical in face recognition or non-rigid which causes 3D inconsistency, we conduct corresponding experiments in the object detection scenario to evaluate the generalizability of EAD across different shapes of the patch. Specifically, we evaluate rectangle-trained EAD with adversarial patches of varying shapes, while maintaining a fixed occupancy of the patch area. The shapes used for evaluation are depicted in Figure 13, with their respective adversarial examples presented in Figure 14. We present the iterative defense process in Figure 15.

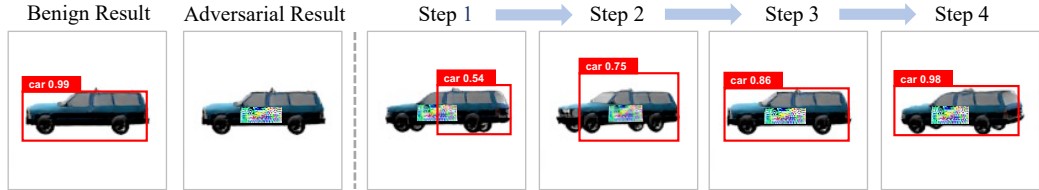

Figure 15: Qualitative results of EAD on simulation environment powered by EG3D. The first two columns present the detection results on be nigh and adversarial images, and the subsequent columns demonstrate the interactive inference steps that the model took. The adversarial patches are generated with MIM and occupy 4% of the image.

As demonstrated in Table 12, our model surpasses other methods in terms of both clean and robust accuracy, even when faced with adversarial patches of diverse unencountered shapes. These results further underscore the exceptional generalizability of EAD in dealing with unknown adversarial attacks.

Table 12: The mAP (%) of YOLOv5 under different white-box adversarial attacks and patch shapes. [†] denotes methods are trained with adversarial examples.

| Method | Clean | Rectangle | | | Ellipse | | | Diamond | | |
|---|---|---|---|---|---|---|---|---|---|---|
| | | PGD | MIM | TIM | PGD | MIM | TIM | PGD | MIM | TIM |
| Undefended | 63.4 | 49.5 | 48.8 | 50.2 | 56.4 | 55.9 | 56.8 | 58.9 | 59.0 | 59.7 |
| JPEG | 62.7 | 51.1 | 50.6 | 57.8 | 57.5 | 57.4 | 57.5 | 60.1 | 59.9 | 59.9 |
| LGS | 64.7 | **64.8** | 55.4 | 60.7 | 65.0 | 66.6 | 61.9 | 65.6 | 66.7 | 63.7 |
| SAC[†] | 63.4 | 49.9 | 47.9 | 48.5 | 52.9 | 51.9 | 52.7 | 57.3 | 56.2 | 56.6 |
| PZ[†] | 63.4 | 50.5 | 50.8 | 50.6 | 56.6 | 57.2 | 57.1 | 59.1 | 59.9 | 59.9 |
| **EAD (ours)** | **75.9** | 64.1 | **58.8** | **67.9** | **70.8** | **67.3** | **72.8** | **71.5** | **69.6** | **74.1** |

