# OpenReview forum: "Embodied Active Defense: Leveraging Recurrent Feedback to Counter Adversarial Patches"
_ICLR.cc/2024/Conference — ICLR 2024 poster_

### Official Review · Reviewer_YXUb · 2023-10-30

**Soundness:** 3 good
**Presentation:** 3 good
**Contribution:** 3 good
**Rating:** 8
**Confidence:** 3

**Summary:**

It develops Embodied Active Defense (EAD), a proactive defensive strategy that actively contextualizes environmental information to address misaligned adversarial patches in 3D real-world settings. To achieve this, EAD develops two central recurrent sub-modules, i.e., a perception module and a policy module, to implement two critical functions of active vision. These models recurrently process a series of beliefs and observations, facilitating progressive refinement of their comprehension of the target object and enabling the development of strategic actions to counter adversarial patches in 3D environments.

**Strengths:**

To optimize learning efficiency, it  incorporates a differentiable approximation of environmental dynamics and deploy patches that are agnostic to the adversary’s strategies.

Extensive experiments demonstrate that EAD substantially enhances robustness against a variety of patches within just a few steps through its action policy in safety-critical tasks (e.g., face recognition and object detection), without compromising standard accuracy.

Furthermore, due to the attack-agnostic characteristic, EAD facilitates excellent generalization to unseen attacks, diminishing the averaged attack success rate by 95% across a range of unseen adversarial attacks.

It theoretically demonstrates the effectiveness of EAD from the perspective of information theory. A well-learned EAD model for contrastive task adopts a greedy informative policy to explore the environment, utilizing the rich context information to reduce the abnormally high uncertainty of scenes caused by adversarial patches.

**Weaknesses:**

It mentions the policy model with actions, and states. It is better to provide more details or examples to specify what the actions and states look like or their physical meanings if any.

It is better to discuss the complexity for the training and inference of the proposed method.

**Questions:**

see the weakness.

---

> ### Author Response · Authors · 2023-11-17
> **Thank you for the valuable review**
>
> We gratefully acknowledge your comprehensive and insightful review of our manuscript. Your detailed feedback and constructive criticism are highly appreciated, as they have significantly contributed to enhancing the overall quality of our work.
>
> We are particularly encouraged by your recognition of the novelty and clarity of our work. In response to your comments and suggestions, we have carefully revised our manuscript. Below, we address each point you raised, aiming to clarify any uncertainties and improve the manuscript further.
>
> 1. **Explanation for the dynamics of the EAD model.**
>
>     To elucidate the actions, states, and their physical meanings in our study, we have added an illustrative example in the new updated Figure 11 (Appendix D.8). These examples demonstrate how the EAD model prioritizes a distinct viewpoint to improve target understanding while trying to keep observing from the viewpoint where adversarial patches are less effective.
>
> 2. **Computational overhead.**
>
>     Thanks for pointing out the complexity of our method compared to the baselines. In response, we have conducted a detailed comparison of computational overhead, encompassing aspects such as parameter amount and computation speed during both training and inference phases. The outcomes of this analysis are presented in Table a.
>
>
>
>      *Table a: Computational overhead comparison of different defense methods in face recognition. We report the training and inference time of defense on a NVIDIA GeForce RTX 3090 Ti and an AMD EPYC 7302 16-Core Processor with the training batch size as 64.*
>
>     | **Method** | **\# Params (M)** | **Description**                                              | **Training Epochs** | **Training Time per batch (s)** | Overall Training time (GPU hours) | **Inference Time per step (ms)** |
>     | ---------- | ----------------- | ------------------------------------------------------------ | ------------------- | ------------------------------- | --------------------------------- | -------------------------------- |
>     | JPEG       | -                 | non-parametric                                               | -                   | -                               | -                                 | 9.65                             |
>     | LGS        | -                 | non-parametric                                               | -                   | -                               | -                                 | 26.22                            |
>     | SAC        | 44.71             | Train a  segmenter to detect a patch area, which consists of two phases: training with pre-generated adversarial images and self-adversarial training | 50 + 10             | 0.152 + 4.018                   | 104                               | 26.43                            |
>     | PZ         | 44.71             | Share the same training procedure and segementer as SAC      | 50 + 10             | 0.152 + 4.018                   | 104                               | 11.88                            |
>     | DOA        | 43.63             | An adversarial training method that retrains the feature extractor. | 100                 | 1.732                           | 376                               | 8.10                             |
>     | EAD (ours) | 57.30             | Train the policy model and perception model in two stages    | 50 + 50             | 0.595 + 1.021                   | 175                               | 11.51                            |
>
>
>
>     As indicated in Table a, in the training phase, the total training time of our EAD model is effectively balanced between the pure adversarial training method DOA and the partially adversarial methods like SAC and PZ. This efficiency stems mainly from our unique USAP approach, which bypasses the need for generating adversarial examples, thereby boosting training efficiency.
>
>      In terms of model inference, our EAD, along with PZ and DOA, demonstrates superior speed compared to LGS and SAC. This is attributed to the latter methods requiring CPU-intensive, rule-based image preprocessing, which diminishes their inference efficiency. Further details, including loss curves and more precise time consumption metrics, are elaborated in Appendix D.6.

---

### Official Review · Reviewer_u8yn · 2023-10-31

**Soundness:** 3 good
**Presentation:** 2 fair
**Contribution:** 3 good
**Rating:** 6
**Confidence:** 2

**Summary:**

In this work, the authors propose an active defense strategy that leverages active movements and recurrent perceptual feedback from the environment to defend against arbitrary adversarial patch attacks. The experiment results show that the proposed strategy can outperform SOTA passive defense strategies in terms of effectiveness and generalizability.

**Strengths:**

1. The authors present a pioneering approach in the realm of adversarial robustness by introducing the first proactive defense strategy with embodied perception. It adds to the current defense strategies against patch attacks, which were predominantly passive.
2. The experiment findings clearly indicate the superiority of the proposed strategy over previous SOTAs. The comprehensive experiments demonstrate that an embodied model with environmental interaction ability can not only mitigate the uncertainty posed by adversarial attacks easily but also be trained with weak attacks like uniform-perturbed patches.

**Weaknesses:**

1. In Section 4.2, the authors mention the effectiveness against adaptive attack; however, I find it hard to understand the reason why a surrogate **uniform** superset policy distribution would necessitate an optimized patch to handle various action policies, as this uniform surrogate may not contain any useful information about the policy model in EAD for the adversary to attack.
2. The paper could benefit from improved clarity, especially for readers with a foundational understanding of adversarial robustness but limited exposure to RL/Robotics. As it stands, the document is dense with jargon, making it challenging to navigate and comprehend upon the initial read.

**Questions:**

1. For those adversarially-trained passive defense models, would it be beneficial to enhance them with 3D-augmented data, e.g., feeding multiple views of the same patch-attacked human face during training, if the lack of 3D environment awareness is the problem here?
2. There is one part I feel confused about in the algorithm box: $O_t′ \gets A(O_t, P; S_t)$ and $O_{t+1}'\gets A(O_{t+1},P;S_{t+1})$, where do $O_t'$ and $O_{t+1}'$ go? They are not explicitly used or referenced anywhere else post-assignment. Are clean observations $O_t$ and $O_{t+1}$ overwritten by $O_t'$ and $O_{t+1}'$ after applied with the adversarial patch by default?

---

> ### Author Response · Authors · 2023-11-17
> **Thank you for the valuable review (1/2)**
>
> We deeply appreciate the time and effort you have invested in reviewing our manuscript. Your insightful feedback has been invaluable in enhancing the quality of our work. We are particularly encouraged by your recognition of the novelty and effectiveness of our proposed active defense strategy.
>
> In this revised submission, we have carefully addressed each of your comments and questions.  Below, we provide detailed responses to your specific points, aiming to clarify the aspects you've raised and further reinforce the strengths of our work.
>
> 1. **Effectiveness of proposed adaptive attack**.
>
>     Your inquiry into the efficacy of a surrogate uniform superset policy distribution is highly relevant. In our initial submission, attacking EAD across a 4-step trajectory was impractical due to excessive GPU memory requirements (approximately 90 GB for 4 steps). Besides our original technique, we have explored gradient checkpointing [1], a method that allows for recalculating parts of the computation graph established by the $\tau$-step EAD inference process, rather than storing them directly. This reduces memory usage while increasing computational demands. Employing this method, we successfully attacked the entire pipeline along a 4-step trajectory using an NVIDIA RTX 3090 Ti.
>
>     To empirically assess the Uniform Superset Policy (USP), we compared it with our new adaptive attack leveraging gradient checkpointing (denoted as GC) across the entire trajectory.
>
>
>
>     *Table a: Evaluation of Adaptive Attacks on the EAD Model. Columns marked **USP** display results from optimizing the patch using an expected gradient over USP. **GC** columns represent results from attacking EAD following the gradients along the 4-step trajectory with gradient checkpointing.*
>
>     |              | **Dodging** |        | **Impersonation** |        |
>     | :----------: | :---------: | :----: | :---------------: | :----: |
>     |              |   **USP**   | **GC** |      **USP**      | **GC** |
>     | **ASR (\%)** |  **22.11**  | 15.79  |       **8.33**        |  7.29  |
>
>
>
>     As Table a demonstrates, our original adaptive attack using USP was more effective than tracing the authentic policy of EAD. This may be attributed to vanishing or exploding gradients impeding optimization [2]. This problem is potentially mitigated by our approach that computes expectations over a uniform policy distribution. Additionally, our method maintains its effectiveness against various adaptive attacks. This underscores that EAD's defense stems from the synergistic integration of its policy and perception models, encouraging strategic data collection rather than merely countering adversarial patches with specific viewpoints. Further details and evaluation results on these adaptive attacks are elaborated in the revised Appendix D.3 and D.7 respectively.
>
> 2. **Clarity improvement**.
>
>     Thanks for your pointing it out. Regarding your concerns about clarity for readers with foundational knowledge of adversarial robustness but limited exposure to RL/Robotics, we have added a section in simplified language to introduce the relevant background, enhancing readability. The comprehensive background information is detailed in the Appendix B. We hope that these revisions make our work more accessible to a broader audience.

---

> > ### Author Response · Authors · 2023-11-17
> > **Thank you for the valuable review (2/2)**
> >
> > 3. **Enhancing passive defense models with 3D-augmented data.**
> >
> >     This is a potential misunderstanding, and we humbly clarify that we have already incorporated 3D augmented images to train the passive baseline, aiming to optimize baseline performance for a fair comparison. Additionally, we conducted training exclusively with 2D data. We have summarized the performance of baselines trained with 2D data (SAC-2D, PZ-2D), 3D augmented data (SAC, PZ), and our EAD model respectively.
> >
> >
> >
> >     *Table b: Performance comparison of passive baseline with different types of data.*
> >
> >     | Method         | Acc. (\%) | Dodging ASR (\%) |         |         |          |          | Impersonation ASR (\%) |         |         |         |         |
> >     | -------------- | --------: | ---------------: | ------: | ------: | -------: | -------: | ---------------------: | ------: | ------: | ------: | :-----: |
> >     |                |           |              MIM |     EoT |   GenAP |    3DAdv |     Adpt |                    MIM |     EoT |   GenAP |   3DAdv |  Adpt   |
> >     | *SAC-2D*       |   *86.22* |           *74.5* |  *82.9* |  *81.9* |   *75.5* |   *48.9* |                 *45.0* |  *49.0* |  *68.0* |  *61.0* |  *N/A*  |
> >     | *PZ-2D*        |   *88.78* |           *61.1* |  *66.7* |  *85.6* |   *76.7* |     74.0 |                 *48.5* |  *51.5* |  *84.8* |  *75.8* | *69.7*  |
> >     | SAC            |     80.55 |             78.8 |    78.6 |    79.6 |     85.8 |     85.0 |                    3.2 |     3.2 |    18.9 |    22.1 |  51.7   |
> >     | PZ             |     85.85 |              6.1 |     6.2 |    14.3 |     20.4 |     69.4 |                **3.1** |     3.2 |    19.1 |    27.4 |  61.0   |
> >     | **EAD (ours)** | **90.45** |          **0.0** | **0.0** | **2.1** | **13.7** | **22.1** |                    4.1 | **3.1** | **5.1** | **7.2** | **8.3** |
> >
> >
> >
> >     As illustrated in Table b, our EAD model outperforms the others, and the 3D-augmented passive baselines are more robust than those trained solely with 2D data. This suggests that enhanced 3D environmental awareness improves adversarial robustness. Furthermore, the active policy of EAD effectively utilizes 3D environmental information to defend against adversarial patches, resulting in superior performance.
> >
> >
> > 4. **Typo in the algorithm box.**
> >
> >     We are so sorry for the typographical errors in the algorithm box, which may have led to confusion. Your interpretation regarding the notation is correct; the perturbed observations $O'$ were intended for training, but we inadvertently omitted the $'$. We have amended these errors in the revised paper.
> >
> > In conclusion, we are immensely grateful for your constructive feedback, which has been pivotal in refining our paper. We hope that our revisions can address your concerns and improve the quality of our paper.
> >
> > **References**:
> >
> > [1] Chen, T., Xu, B., Zhang, C., \& Guestrin, C. (2016). Training deep nets with sublinear memory cost. *arXiv preprint arXiv:1604.06174*.
> >
> > [2] Athalye, A., Carlini, N., \& Wagner, D. (2018). Obfuscated Gradients Give a False Sense of Security: Circumventing Defenses to Adversarial Examples. *arXiv:1802.00420 [Cs]*. http://arxiv.org/abs/1802.00420

---

### Official Review · Reviewer_V4oi · 2023-11-04

**Soundness:** 3 good
**Presentation:** 2 fair
**Contribution:** 2 fair
**Rating:** 5
**Confidence:** 3

**Summary:**

The paper proposes a model robustness approach Embodied Active Defense (EAD) for perception models against adversarial patches. EAD comprises of perception module to extract the image observation features and policy module to interact with the environment (modeled by decision transformer). The paper hypothesizes that passive robustness methods (i.e without temporal information or scene contextual information) would not be sufficient for unseen adversarial attacks and needs active feedback from the environment to achieve better model robustness. EAD training is a two stage learning approach performed in a dynamic environment (i.e. the generative space of faces using EG3D or CARLA simulation)  - 1) training the perceptual model for the specific task (for eg face recognition or object detection) using random policy 2) co-training the perception module along with the policy. The paper shows robustness to different types of patch based adversarial attacks and adaptive adversarial attacks. The paper also proposes to introduce uniformly sampled noise as adversarial examples while training the model that perturbs the observation. The paper empirically shows adversarial robustness improvement across unseen type of attacks and also shows improvement on clean samples (i.e overall test accuracy of a task in scenarios where adversarial samples are not introduced).

**Strengths:**

The paper explores an adversarial robustness technique that helps the model to achieve better adversarial robustness in a dynamic environment. The proposed approach “Embodied Active Defense (EAD)” achieves improvement on various patch adversarial attack methods for the tasks of face recognition and object detection. The paper provides ablation experiments to analyze the various components of EAD algorithm. Ablation experiments regarding the strength of adversarial attacks (by increasing the patch size or increasing the iteration of iterative adversarial attacks) sheds light on the robustness of the proposed model against strong version of adversarial attacks. The supplementary material contains code for reproducibility

**Weaknesses:**

Overall, the paper writing style and organization needs improvement to allow the reader to easily understand the paper.
It would be helpful for the reader to get a better understanding for the following:
1.  Some general suggestions regarding paper organizing. It would be great to introduce the problem statement for each of the tasks:
    1.  For eg. definition/explaining of the subtasks of FR : Impersonation and dodging.
    2.  Environmental model definition (State, Action, Transition Function and Observation Function similar to Face Recognition task) for object detection on CARLA simulator or ShapeNet EG3D.
    3. In the related section, a brief explanation of the attacks used for evaluation.
    4. Annotation definitions before introducing in the paper (for eg y^{_}_t prediction of the model)
    5. (Optional) a readme for the codebase, to understand the outline of the codebase.
2. The paper makes a claim that passive adversarial defense approach are not sufficient for dynamics environment. This claim should be supported by a passive defense used as a baseline. Also, it would be great to see the extra amount of training resources used to train the active embodied model vs a passive defense approach (for eg Madry's PGD adversarial training). It would also help the readers if some other active defenses would be used as a baseline in order to establish the efficacy of the proposed approach.
3. The experimental (both training and evaluation) setting does not seem sufficient and scalable enough to make conclusions that it would general. For example “To train EAD for face recognition, we randomly sample images from 2, 500 distinct identities from the training set of CelebA-3D.” and “we report the white-box attack success rate (ASR) on 100 identity pairs in both impersonation and dodging attacks with various attack methods”.

Minor Typo/ suggestions
EAD comprises two primary submodules: -> EAD comprises of two primary submodules:

Notely, it maintains or even improves standard accuracy due to -> Notably, it maintains or even improves standard accuracy due to

Formally, It derives a strategic action -> Formally, it derives

“they have now developed to perceive various perception models”

“It is noteworthy this presents a more versatile 3D formulation for adversarial patches”

from given scene x with its annotation y from another modal like CLIP -> from given scene x with its annotation y from another model like CLIP

“In D.5 MORE EVALUATION RESULTS ON EG3D, We present the iterative defense process in Figure ??.”

In the statement “EAD presents a promising direction for enhancing robustness without any negative social impact”, it might be helpful to limit this statement as the one of the tasks being used is facial recognition that could have some unwanted impact .

**Questions:**

It would be helpful if the paper could answer/clarify the following questions:
1. How much training time/ wall clock time and memory resources does it take for the EAD defense (both at training and inference phase).
2. The general assumption about adversary is that it is not bound by computation resources. For example the statement in the paper for adaptive attacks “While the deterministic and differential approximation could enable backpropagation through the entire inference trajectory of EAD, the computational cost is prohibitive due to rapid GPU memory consumption as trajectory length τ increases”, here for Face recognition can we do a white box attack by following the gradients for all the 4 -step trajectory?
3. In the paper the description for Figure 3 mentions that “subsequent active interactions with the environment progressively reduce the similarity between the positive pair”. Is this a typo, should this be “reduce the dissimilarity” or “increase the similarity”?
4. In Algorithm 1, the perturbed observation O’t is not being used, is there a typo there ?

---

> ### Author Response · Authors · 2023-11-17
> **Thank you for the valuable review (1/3)**
>
> We express our sincere appreciation for your meticulous and insightful review of our submission. Your detailed analysis and constructive feedback are invaluable to us, and we appreciate the time and effort you invested in evaluating our work.
>
> In response to your suggestions and critiques, we have diligently worked on revising our paper to enhance its clarity, organization, and overall contribution to the field. Below, we outline the specific changes and improvements made in accordance with your valuable feedback.
>
> 1. **Improvement of writing and clarity.**
>
>     We understand your concern regarding the paper's writing style and organization. We have worked diligently to address these issues to make readers easily grasp the content. Specifically, we have made the following improvements:
>
>     1.1 A detailed introduction to the tasks of impersonation and dodging attacks has been added in the updated Appendix D.2 for improved clarity.
>
>     1.2 The environmental setup for object detection, similar to that used in face recognition, primarily varies in the feasible region of viewpoints. We've included a comprehensive description of this setting, incorporating the CARLA simulator and ShapeNet EG3D in the tail of Appendix C.
>
>     1.3 The explanation for the used attacks including MIM, EoT, GenAP and Face3DAdv, has been elaborated in  Appendix D.2 for enhanced clarity.
>
>     1.4 Annotation definitions and terms including model prediction are now more clearly defined in the revised methodology section (Sec. 3.1).
>
>     1.5 The codebase outline and a detailed description have been provided to ensure reproducibility.
>
> 2. **Evidence supporting the inadequacy of passive defenses in dynamic environments and baseline concerns**.
>
>     2.1 **Passive defenses are inadequate in dynamic environments**.
>
>     We want to humbly point out that we have incorporated various passive defenses in our submission, including Preprocessed methods (e.g., JPEG Compression, Local Gradients Smoothing (LGS), Segment and Complete (SAC) and PatchZero (PZ)) alongside an adversarial training based method for patch attacks, namely Defense against Occlusion Attacks (DOA), which is similar to Madry's PGD adversarial training. These methods represent common and state-of-the-art passive defenses against patch attacks. The comparative results are detailed in Table 1 and Figure 2 of our original submission.
>
>     2.2 **Extra training resources**.
>
>     Regarding training resources, our method demonstrates an obviously reduced training overhead compared to traditional adversarial training approaches, such as DOA. Adversarial training typically incurs substantially higher computational costs than standard training methods [1], particularly when designed to counter a broad range of attacks. Despite our model necessitating an additional policy model and a temporal fusion model, its training resource requirements remain considerably lower compared to those of adversarial training-based methods (e.g., DOA). For the training of EAD only requires patches from Uniform Superset Approximation thereby bypassing the expensive computational cost of inner maximization of adversarial training. Furthermore, our approach is orthogonal to adversarial training, making it feasible to integrate the latter to enhance EAD's resilience against specific attacks. We also provide detailed discussions on computational resources below in "Computational overhead for EAD defense".
>
>     2.3 **Active defense baselines.**
>
>     To the best of our knowledge, our work represents the first attempt to promote the adversarial robustness of perception in an active paradigm.
>
> 3. **Sufficiency and scalability of experimental setting.**
>
>     As for the data sufficiency in our experimental settings,  we would like to clarify that our training dataset encompasses over 3M unique images, rendered from various 3D facial models and viewpoints. Each identity has approximately 20 distinct 3D facial models, with nearly 60 images rendered from each model. This quantity of face data is exactly the same as previous adversarial defense work like SAC [2]. Besides, this amount has reached the average volume of current top-tier FR dataset (e.g., 1.2M for Digi-Face 1M [3], 3.3M for VGG Face2 [4], 0.5M for CASIA-WebFace [5]). As for testing, we adhere to the standard testing protocol in [6]. The testing dataset is equally abundant and diverse with ones in the training phase, which involves various 3D facial models and viewpoints. Furthermore, we are also generating additional data to further validate the scalability and generalizability of our findings, with plans to update our findings accordingly.

---

> > ### Author Response · Authors · 2023-11-17
> > **Thank you for the valuable review (2/3)**
> >
> > 4. **Minor typos/suggestions.**
> >
> >     We deeply apologize for any ambiguity or errors in our initial submission and have meticulously corrected these in the revised version. We have thoroughly refined the entire paper, paying special attention to clarity and precision in our descriptions. Additionally, following your advice, we have reevaluated the ethical implications of our work. In light of this, we have carefully modified relevant statements to more accurately reflect the potential impacts of our research, ensuring a more responsible presentation.
> >
> > 5. **Computational overhead for EAD defense.**
> >
> >     Thanks for pointing out the computational cost of our method compared to the baselines. In response, we have conducted a detailed comparison of computational overhead, including parameter amount and computation speed, in both training and inference phases, summarized in the following table:
> >
> >
> >
> >     *Table a: Computational overhead comparison of different defense methods in face recognition. We report the training and inference time of defense on a NVIDIA GeForce RTX 3090 Ti and an AMD EPYC 7302 16-Core Processor with the training batch size as 64.*
> >
> >     | **Method** | **\# Params (M)** | **Description**                                              | **Training Epochs** | **Training Time per batch (s)** | Overall Training time (GPU hours) | **Inference Time per step (ms)** |
> >     | ---------- | ----------------- | ------------------------------------------------------------ | ------------------- | ------------------------------- | --------------------------------- | -------------------------------- |
> >     | JPEG       | -                 | non-parametric                                               | -                   | -                               | -                                 | 9.65                             |
> >     | LGS        | -                 | non-parametric                                               | -                   | -                               | -                                 | 26.22                            |
> >     | SAC        | 44.71             | Train a  segmenter to detect a patch area, which consists of two phases: training with pre-generated adversarial images and self-adversarial training | 50 + 10             | 0.152 + 4.018                   | 104                               | 26.43                            |
> >     | PZ         | 44.71             | Share the same training procedure and segmenter as SAC       | 50 + 10             | 0.152 + 4.018                   | 104                               | 11.88                            |
> >     | DOA        | 43.63             | An adversarial training method that retrains the feature extractor. | 100                 | 1.732                           | 376                               | 8.10                             |
> >     | EAD (ours) | 57.30             | Train the policy model and perception model in two stages    | 50 + 50             | 0.595 + 1.021                   | 175                               | 11.51                            |
> >
> >
> >
> >     As indicated in Table a, in the training phase, the total training time of our EAD model is effectively balanced between the pure adversarial training method DOA and the partially adversarial methods like SAC and PZ. This efficiency stems mainly from our unique USAP approach, which bypasses the need for generating adversarial examples, thereby boosting training efficiency.
> >         In terms of model inference, our EAD, along with PZ and DOA, demonstrates superior speed compared to LGS and SAC. This is attributed to the latter methods requiring CPU-intensive, rule-based image preprocessing, which diminishes their inference efficiency. Further details, including loss curves and more precise time consumption metrics, are elaborated in Appendix D.6.

---

> > > ### Author Response · Authors · 2023-11-17
> > > **Thank you for the valuable review (3/3)**
> > >
> > > 6. **Computational overhead for adaptive attack.**
> > >
> > >     In our initial submission,  attacking EAD by following the gradients for all the 4-step trajectory is prohibitive due to the substantial GPU memory consumption  (4 steps require over 90 GB video memory). Besides the technique presented in our submission, we further investigate a new technique, namely gradient checkpointing [7]. By checkpointing nodes in the computation graph defined by the $\tau$-step EAD inference procedure, this technique recomputes the parts of the graph instead of directly storing them, thereby reducing memory cost with extra computation. With this technique, we succeed in attacking the whole pipeline along 4-step trajectory (overall) on a NVIDIA RTX 3090 Ti.
> > >
> > >     We conduct the adaptive attack by following the gradients along the whole trajectory with gradient checkpointing technique (denoted by GC) and show the results along with the adaptive attack with surrogate Uniform Superset Policy (denoted by USP) which is proposed in our previous submission.
> > >
> > >    *Table b: Evaluation of adaptive attacks on the EAD Model. Columns with **USP** represent results obtained by optimizing the patch with an expected gradient over the Uniform Superset Policy (USP). And **GC** denotes attacking EAD by following the gradients for along the 4-step trajectory with gradient checkpointing.*
> > >
> > >     |              | **Dodging** |        | **Impersonation** |        |
> > >     | :----------: | :---------: | :----: | :---------------: | :----: |
> > >     |              |   **USP**   | **GC** |      **USP**      | **GC** |
> > >     | **ASR (\%)** |  **22.11**  | 15.79  |       8.33        |  7.29  |
> > >
> > >     As depicted from the above results,  EAD stills maintains its robustness when faced adaptive attack against its whole pipeline. It further shows that EAD’s defensive capabilities arise from the synergistic integration of its policy and perception models, facilitating strategic observation collection rather than learning a short-cut strategy to neutralize adversarial patches from specific viewpoints. Moreover, the detailed methodologies of these adaptive attacks and relevant results are now illustrated in Appendix D.3 and D.7 respectively.
> > >
> > > 7. **Question about the description of Figure 3.**
> > >
> > >     Thank you for your insightful query regarding the description of Figure 3 in our paper. Upon reviewing your question, we have identified that there is a typographical error in our manuscript. The correct phrasing should be “reduce the dissimilarity” or, equivalently, “increase the similarity” between the positive pair. We have corrected this typographical error, aligning it with the standards of academic excellence.
> > >
> > > 8. **Question about the notation in Algorithm 1.**
> > >
> > >     Thank you for your attentive reading of our manuscript and for highlighting the concern regarding Algorithm 1. Upon revisiting the algorithm, this is indeed a typographical error in our paper.
> > >
> > >     To clarify, $O_t'$ should be integrated into the algorithm at 7th and 9th line, where it plays a crucial role in building patch-countering capability of the EAD. We appreciate your keen eye in identifying this oversight, and we will ensure that this correction is made to accurately represent the workings of the algorithm in the revised manuscript.
> > >
> > > Your constructive feedback has been instrumental in refining our research. We hope that our responses and the modifications can address your concerns.
> > >
> > > **References:**
> > >
> > > [1] Wong, E., Rice, L., & Kolter, J. Z. (2020). Fast is better than free: Revisiting adversarial training. arXiv preprint arXiv:2001.03994.
> > >
> > > [2] Liu, J., Levine, A., Lau, C. P., Chellappa, R., \& Feizi, S. (2022). Segment and complete: Defending object detectors against adversarial patch attacks with robust patch detection. In *Proceedings of the IEEE/CVF Conference on Computer Vision and Pattern Recognition* (pp. 14973-14982).
> > >
> > > [3] Bae, G., de La Gorce, M., Baltrušaitis, T., Hewitt, C., Chen, D., Valentin, J., ... \& Shen, J. (2023). Digiface-1m: 1 million digital face images for face recognition. In *Proceedings of the IEEE/CVF Winter Conference on Applications of Computer Vision* (pp. 3526-3535).
> > >
> > > [4] Cao, Q., Shen, L., Xie, W., Parkhi, O. M., \& Zisserman, A. (2018, May). Vggface2: A dataset for recognizing faces across pose and age. In *2018 13th IEEE international conference on automatic face \& gesture recognition (FG 2018)* (pp. 67-74). IEEE.
> > >
> > > [5] Yi, D., Lei, Z., Liao, S., \& Li, S. Z. (2014). Learning face representation from scratch. *arXiv preprint arXiv:1411.7923*.
> > >
> > > [6] Liu, J., Levine, A., Lau, C. P., Chellappa, R., \& Feizi, S. (2022). Segment and complete: Defending object detectors against adversarial patch attacks with robust patch detection. In *Proceedings of the IEEE/CVF Conference on Computer Vision and Pattern Recognition* (pp. 14973-14982).
> > >
> > > [7] Chen, T., Xu, B., Zhang, C., \& Guestrin, C. (2016). Training deep nets with sublinear memory cost. *arXiv preprint arXiv:1604.06174*.

---

> ### Author Response · Authors · 2023-11-20
> **Look forward to further feedback**
>
> Dear Reviewer V4oi,
>
> We thank you again for the valuable and insightful comments. We hope you might find the response satisfactory and are looking forward to hearing from you about any further feedback.
>
> Best, Authors

---

> ### Comment · Reviewer_V4oi · 2023-11-22
> **Thank you for the response**
>
> I really appreciate the author's explanations to my concerns. These discussions mostly answers my questions and the revised manuscript seems to have better readability. Accordingly I have increased my overall rating to 5 (and increased the Soundness and Presentation score).

---

> > ### Author Response · Authors · 2023-11-22
> > **Thanks for the update**
> >
> > Thank you very much for increasing the rating and valuable comments. We'll try our best to further improve the paper in the final version.

---

### Official Review · Reviewer_iMPU · 2023-11-05

**Soundness:** 3 good
**Presentation:** 4 excellent
**Contribution:** 4 excellent
**Rating:** 8
**Confidence:** 3

**Summary:**

The paper proposes a defense against patch-based adversarial attacks on visual recognition models. It utilizes the concept of emodied perception (wherein the model is allowed to interact with its environment to refine its belief) to mitigate the effect of 'out-of-place' patches in the scene and, in turn, improve the model's robustness against patch-based attacks. The authors implement embodied perception as a partially observable markov decision process. The overall system comprises of two new models: a perception model and a policy model. The perception model maintains an 'understanding' of the scene which it progressively updates based on new observations. The policy model dictates a transition process that is focused on improving the quality of observations to improve recognition. Given an initial observation, the system progressively refines its belief using the perception and policy models. Through two visual recognition tasks: face recognition and object detection, authors demonstrate the effectiveness of emodied perception based defense to not only improve robustness against seen and unseen attacks, but also improve standard performance.

**Strengths:**

1. Overall, the writing quality and presentation of the paper is excellent. The authors lay out the motivation behind their method and challenges behind implementing it adequately. All necessary background is provided so that a person unfamiliar with the topic can follow along. The figures and tables present information effectively.
2. The authors propose a novel application of emodied perception, using it as a defense against patch-based adversarial attacks, making it the first defense of its kind.
3. The design of the method is clever and intricate, and is intuitively grounded in the understanding of human perception.
4. The defense is task agnostic, and should conceptually work with any visual recognition task.
5. The authors provide a theoretical understanding for the effectiveness of EAD using information theory.
6. The ablation presented in the paper is thorough, and covers all the different components of the method as well as newly introduced hyperparameters.
7. The method improves both standard and adversarial performance relative to prior defenses. Interestingly, standard performance is improved even over undefended baseline. Overall, improvements introduced by proposed method appear substantial.

**Weaknesses:**

1. There is no discussion regarding the computational cost of the proposed method and how it compares to the undefended baseline as well as prior defenses.
2. An end-to-end attack is not necessarily the strongest adaptive attack, especially for defenses with complicated forward passes. Attacking the weakest component should be sufficient. To identify the weakest component, authors should try independently attacking perception and policy models to make them less effective in their respective tasks. For example, attacking perception model would involve generating an input that forces the perception model to output a corrupted internal belief vector b_t (using a latent space attack [a]).
3. There are no theoretical results establishing how the approximations used in Sec 3.3 relate to the actual quantities. If possible to obtain, this would be nice to have.

**References**

[a] Sabour, S., Cao, Y., Faghri, F., and Fleet, D. J. Adversarial manipulation of deep representations. International Conference on Learning Representations, 2016.

**Questions:**

1. How does the computational cost of the proposed method compares to the baselines?
2. How does the method fare against an adaptive attack targeting the weakest component in the pipeline (see further details in the Weaknesses section)? I'd love to increase my rating further if you can better convince me that the defense is robust against adaptive attacks.
3. Why is the enhanced version of SAC less robust against dodging attacks than regular SAC (in table 1)?

---

> ### Author Response · Authors · 2023-11-17
> **Thank you for the valuable review (1/2)**
>
> We sincerely appreciate your detailed and constructive feedback. Your insights have greatly enriched our understanding and presentation of the research, enabling us to refine and strengthen our work significantly. We are particularly grateful for your recognition of the novelty and clarity of our approach, as well as the thoroughness of our experiments.
>
> In response to your valuable comments, we have meticulously revised our paper to address each point you raised. We hope that our revisions and responses meet your expectations.
> Below, we provide detailed responses to each of the points you have raised:
>
> 1. **Discussion over computational overhead**
>
>     Thanks for pointing out the computational cost of our method compared to the baselines. In response, we have conducted a detailed comparison of computational overhead, encompassing aspects such as parameter amount and computation speed during both training and inference phases. The outcomes of this analysis are presented in Table a.
>
>
>
>     *Table a: Computational overhead comparison of different defense methods in face recognition. We report the training and inference time of defense on a NVIDIA GeForce RTX 3090 Ti and an AMD EPYC 7302 16-Core Processor with the training batch size as 64.*
>
>     | **Method** | **\# Params (M)** | **Description**                                              | **Training Epochs** | **Training Time per batch (s)** | Overall Training time (GPU hours) | **Inference Time per step (ms)** |
>     | ---------- | ----------------- | ------------------------------------------------------------ | ------------------- | ------------------------------- | --------------------------------- | -------------------------------- |
>     | JPEG       | -                 | non-parametric                                               | -                   | -                               | -                                 | 9.65                             |
>     | LGS        | -                 | non-parametric                                               | -                   | -                               | -                                 | 26.22                            |
>     | SAC        | 44.71             | Train a segmenter to detect a patch area, which consists of two phases: training with pre-generated adversarial images and self-adversarial training | 50 + 10             | 0.152 + 4.018                   | 104                               | 26.43                            |
>     | PZ         | 44.71             | Share the same training procedure and segmenter as SAC       | 50 + 10             | 0.152 + 4.018                   | 104                               | 11.88                            |
>     | DOA        | 43.63             | An adversarial training method that retrains the feature extractor | 100                 | 1.732                           | 376                               | 8.10                             |
>     | EAD (ours) | 57.30             | Train the policy model and perception model in two stages    | 50 + 50             | 0.595 + 1.021                   | 175                               | 11.51                            |
>
>
>
>     As indicated in Table a, in the training phase, the total training time of our EAD model is effectively balanced between the pure adversarial training method DOA and the partially adversarial methods like SAC and PZ. This efficiency stems mainly from our unique USAP approach, which bypasses the need for generating adversarial examples, thereby boosting training efficiency.
>     In terms of model inference, our EAD, along with PZ and DOA, demonstrates superior speed compared to LGS and SAC. This is attributed to the latter methods requiring CPU-intensive, rule-based image preprocessing, which diminishes their inference efficiency. Further details, including loss curves and more precise time consumption metrics, are elaborated in Appendix D.6.

---

> ### Author Response · Authors · 2023-11-17
> **Thank you for the valuable review (2/2)**
>
> 2. **More adaptive attacks.**
>
>     Thanks for pointing it out. We have now involved three new adaptive attacks to assess the robustness of the EAD model. These attacks target different components: the perception model, the policy model, and the entire pipeline.
>
>     -  **Perception:** Attacking the perception model to disrupt the model belief (perception) according to your advice and the work by Sabour et al. [1]
>     -  **Policy:** Attacking the policy model to make it ensure zero action (i.e., EAD model stay stills) and output erroneous predictions.
>     -  **Overall:** In our previous version,  attacking EAD by backpropagation is prohibitive due to rapid GPU memory consumption as trajectory length increases (4 steps require over 90 GB video memory). With further investigation, we find a new technique to address that issue, namely gradient checkpointing [2]. By checkpointing nodes in the computation graph defined by the $\tau$-step EAD inference procedure, this technique recomputes the parts of the graph instead of directly storing them, thereby reducing memory cost with extra computation. With this technique, we succeed in attacking the whole pipeline along 4-step trajectory on a NVIDIA RTX 3090 Ti.
>
>     We summarize the results in Table b.
>
>
>
>     *Table b: Evaluation of adaptive attacks on the EAD Model. Columns with **USP** represent results obtained by optimizing the patch with an expected gradient over the Uniform Superset Policy (USP). And **perception** and **policy** separately represent adaptive attacks against a single sub-module. And **overall** denotes attacking EAD by following the gradients for along the overall (4 steps) trajectory with gradient checkpointing.*
>
>     |              | **Dodging** |                |            |             | **Impersonation** |                |            |             |
>     | :----------: | ----------- | :------------: | :--------: | :---------: | :---------------: | :------------: | :--------: | :---------: |
>     |              | **USP**     | **Perception** | **Policy** | **Overall** |      **USP**      | **Perception** | **Policy** | **Overall** |
>     | **ASR (\%)** | **22.11**   |     10.11      |   16.84    |    15.79    |       8.33        |      1.04      |  **9.38**  |    7.29     |
>
>
>
>     The results further validate the robustness of EAD against a spectrum of adaptive attacks. It further shows that EAD’s defensive capabilities arise from the synergistic integration of its policy and perception models, facilitating strategic observation collection rather than learning a short-cut strategy to neutralize adversarial patches from specific viewpoints.
>     The methodologies and detailed results of these adaptive attacks are now thoroughly documented in Appendix D.3 and Appendix D.7 respectively.
>
> 3. **Theoretical results relating to approximations in Sec 3.3.**
>
>     Thanks for your suggestion. We agree that while our method is fundamentally grounded in information theory, there are several approximations in our implementation that merit deeper theoretical exploration. We would like to further achieve this as a potential area for future work, aiming to provide a more comprehensive theoretical analysis that aligns closely with our practical implementations.
>
> 4. **Enhanced SAC is less robust than the regular one against dodging attacks.**
>
>     SAC is a preprocessing-based method that adopts a segmentation model to detect patch areas, followed by a "shape completion" technique to extend the predicted area into a larger square, and remove the suspicious area. We observed that the enhanced SAC, while exhibiting superior segmentation performance, inadvertently increases the likelihood of masking critical facial features such as eyes and noses in scenarios like face recognition. As a result, the Face Recognition (FR) model is more likely to misidentify the masked faces as different identities, thereby reducing the efficacy of dodging attacks. Besides, we also provide an illustrative example (Figure 10) and more details in the Appendix D.8 to elucidate this phenomenon.
>
> **References:**
>
> [1] Sabour, S., Cao, Y., Faghri, F., \& Fleet, D. J. (2015). Adversarial manipulation of deep representations. *arXiv preprint arXiv:1511.05122*.
>
> [2] Chen, T., Xu, B., Zhang, C., \& Guestrin, C. (2016). Training deep nets with sublinear memory cost. *arXiv preprint arXiv:1604.06174*.

---

### Author Response · Authors · 2023-11-22
**Look forward to further feedback**

Dear reviewers,

We sincerely appreciate your insightful comments and the time you have dedicated to reviewing our work. We are looking forward to hearing from you about any further feedback.

If our response meets your expectations, we hope that you might view this as a sufficient reason to further raise your score.

If you still have any further questions regarding our paper, we are dedicated to discussing them with you and improving our paper.

Best, Authors

---

### Meta-Review · Area_Chair_c29t · 2023-12-11

**Metareview:**

The paper proposes a defense against patch-based adversarial attacks on visual recognition models. It utilizes the concept of emodied perception (wherein the model is allowed to interact with its environment to refine its belief) to mitigate the effect of 'out-of-place' patches in the scene and, in turn, improve the model's robustness against patch-based attacks. The overall system comprises of two new models: a perception model and a policy model. The perception model maintains an 'understanding' of the scene which it progressively updates based on new observations. The policy model dictates a transition process that is focused on improving the quality of observations to improve recognition. Given an initial observation, the system progressively refines its belief using the perception and policy models.

The authors have effectively addressed most of the reviewers' concerns during the rebuttal phase. While the scope of defense against adversarial patch attacks is limited, the proposed method stands as a novel approach, providing insights into adversarial defense strategies and exhibiting effectiveness in experiments.

In all, we recommend accepting this work to ICLR and hope the authors take the reviewers' suggestions into their final version.

**Justification For Why Not Higher Score:**

Defense against adversarial patch attacks is limited in scope.

**Justification For Why Not Lower Score:**

Sufficient novelty, insights, and effectiveness of the methods demonstrated in experiments.

---

### Decision · Program_Chairs · 2024-01-16

Accept (poster)